# Recent inorganic carbon increase in a temperate estuary driven by water quality improvement and enhanced by droughts

Louise C. V. Rewrie[1*], Burkard Baschek[2], Justus E. E. van Beusekom[1], Arne Körtzinger[3], Gregor Ollesch[4], Yoana G. Voynova[1]

1. Institute of Carbon Cycles, Helmholtz-Zentrum Hereon, 21502 Geesthacht, Germany.
2. Deutsches Meeresmuseum, 18439 Stralsund, Germany
3. GEOMAR Helmholtz-Zentrum für Ozeanforschung Kiel, 24148 Kiel, Germany
4. Flussgebietsgemeinschaft Elbe (FGG Elbe), 39104 Magdeburg, Germany

*Correspondence to:* Louise C. V. Rewrie (louise.rewrie@hereon.de) and Yoana G. Voynova (yoana.voynova@hereon.de).

**Abstract**

Estuaries are an important contributor to the global carbon budget, facilitating carbon removal, transfer and transformation between land and the coastal ocean. Estuaries are susceptible to global climate change and anthropogenic perturbations. We find that a long-term significant mid-estuary increase in dissolved inorganic carbon (DIC) of 6–21 µmol kg$^{-1}$ yr$^{-1}$ (1997–2020) in a temperate estuary in Germany (Elbe Estuary), was driven by an increase in upper estuary particulate organic carbon (POC) content of 8–14 µmol kg$^{-1}$ yr$^{-1}$. The temporal POC increase was due to an overall improvement in water quality observed in the form of high rates of primary production and a significant drop in biological oxygen demand. The magnitude of mid-estuary DIC gain was equivalent to the increased POC production in the upper estuary, suggesting that POC is effectively remineralized and retained as DIC in the mid-estuary, with the estuary acting as an efficient natural filter for POC. In the context of this significant long-term DIC increase, a recent extended drought period (2014–2020) significantly lowered the annual mean river discharge (468 ± 234 m$^3$ s$^{-1}$) compared to the long-term mean (690 ± 441 m$^3$ s$^{-1}$, 1960–2020), while the late spring internal DIC load in the estuary doubled. The drought induced a longer dry season, starting in May (earlier than normal), increased the residence time in the estuary and allowed for a more complete remineralization period of POC. Annually, 77–94 % of the total DIC export was laterally transported to the coastal waters, reaching 89 ± 4.8 Gmol C yr$^{-1}$, and thus, between 1997 and 2020, only an estimated maximum of 23% (10 Gmol C yr$^{-1}$) was released via carbon dioxide (CO$_2$) evasion. Export of DIC to coastal waters decreased significantly during the drought, on average by 24% (2014–2020: 38 ± 5.4 Gmol C yr$^{-1}$), compared to the non-drought period. In contrast, there was no change in the water-air CO$_2$ flux during the drought. We have identified that seasonal changes in DIC processing in an

estuary require consideration when estimating both the long-term and future changes in water-air $CO_2$ flux and DIC export to coastal waters. Regional and global carbon budgets should therefore take into account carbon cycling estimates in estuaries, and their changes over time in relation to impacts of water quality changes and extreme hydrological events.

## **Graphical Abstract**

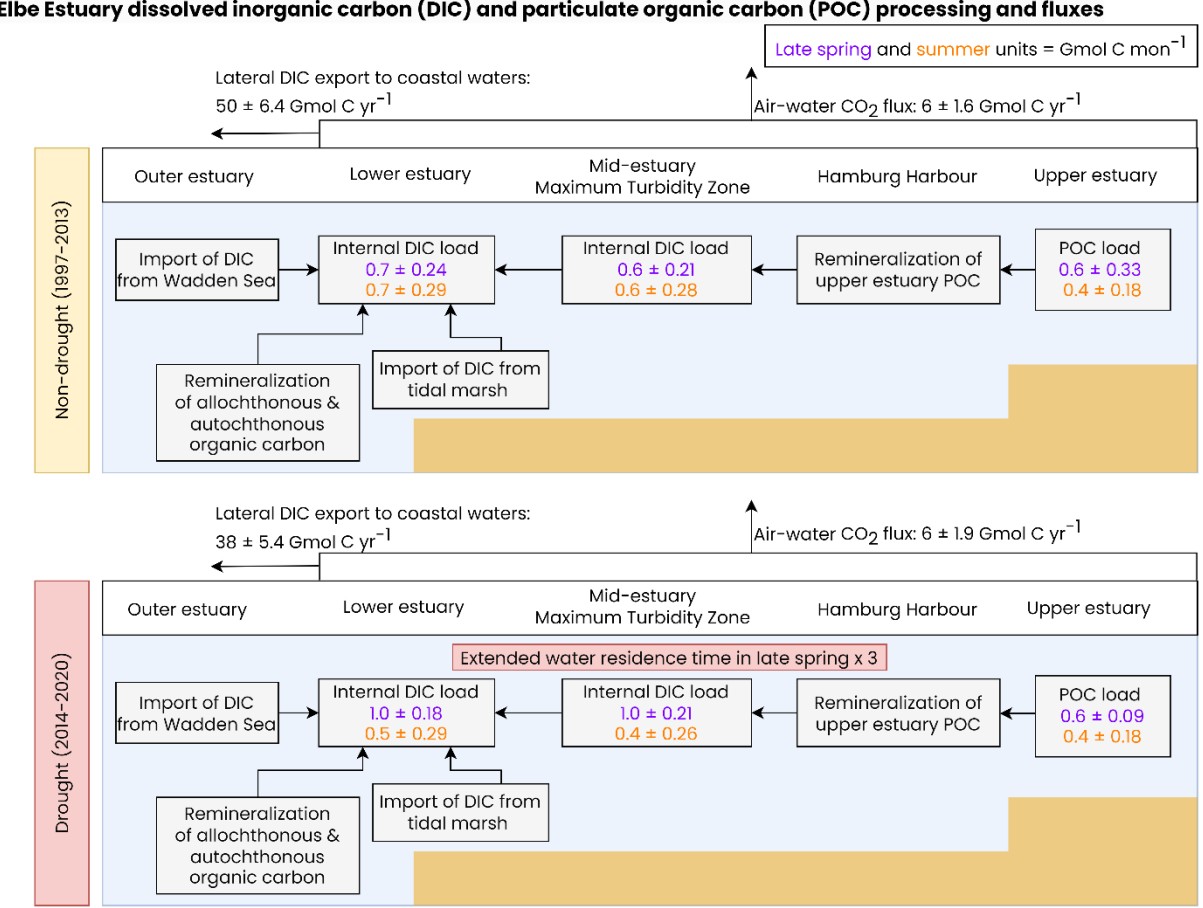

## **1. Introduction**

Estuaries function as bioreactors in which biotic and abiotic processes act to augment, transform or attenuate carbon products (Bukaveckas, 2022). Despite successful initiatives to reduce eutrophication in estuaries and coastal waters, e.g. in the Wadden Sea (van Beusekom et al., 2019) and Delaware Estuary (Sharp, 2010), effects of anthropogenic eutrophication persist (Harding et al., 2019). Rivers still receive large nutrient concentrations sustaining enhanced phytoplankton growth, due to agricultural land use dominating the river catchments such as along the Rhine and Elbe Rivers (Hardenbicker et al., 2016; Dähnke et al. 2022). River-borne

and in situ primary production supplies allochthonous and autochthonous organic carbon to and within estuaries (Abril et al. 2002; Hoellein et al. 2013), subsequently providing labile forms of carbon. This organic matter (OM) input into the estuary can lead to net heterotrophic

conditions in an estuary (Schöl et al. 2014), with OM further decomposed and converted into dissolved inorganic carbon (DIC). Intense respiration of OM in estuaries can elevate the partial pressure of carbon dioxide ($p$CO$_2$) to above atmospheric levels, resulting in estuarine regions acting as a CO$_2$ sources to the atmosphere (Amann et al., 2015; Cai 2011). This reduces labile OM export to the adjacent coastal waters (Abril et al. 2002; Crump et al., 2017; Sanders et al.,

2018). The rate of heterotrophic activity in estuaries, such as bacterial production and respiration, has been shown to correlate with phytoplankton production (Hoch and Kirchman, 1993), POC concentration (Goosen et al., 1999) and temperature (Apple et al., 2006). This highlights the need to understand how changes in primary production of OM in the upstream estuary affect downstream heterotrophic conditions and production and export of DIC.


Over the last century, global temperatures have increased by 0.95 to 1.20°C (IPCC, 2022). Future global temperature increases are projected to intensify the hydrological cycle, while climate projections show that the frequency and length of droughts (Böhnisch et al., 2021), as well as the frequency and magnitude of heavy precipitation and flood events (Christensen and

Christensen, 2003; Alfieri et al., 2015), will increase across Europe. Such modifications in the hydrological balance will influence river systems, which are among the most sensitive ecosystems to climate change (Watts et al., 2015). While extreme floods tend to reduce residence time in estuaries and generate a large export of OM and nutrients from land to coastal waters (Voynova et al., 2017), drought conditions can lengthen river and estuarine water

residence time (Hitchcock and Mitrovic, 2015). This in turn can extend the retention of carbon and nutrients during droughts, permitting more extensive remineralization of allochthonous and autochthonous OM within an estuary (Hitchcock and Mitrovic, 2015), subsequently altering carbon and nutrient cycling. With hydrological droughts predicted to become more frequent and extensive in Europe (Forzieri et al., 2014; Williams et al., 2015), assessing how they

influence carbon dynamics and estuarine biogeochemistry is essential for understanding and predicting carbon storage and export to coastal regions.

This study aims to highlight the functioning of an estuary under a multi-year drought, in the context of current regional climate change predictions. The Elbe Estuary is used as an example

of a temperate estuary with a densely populated watershed, and subject to severe drought

conditions between 2014 and 2020 (Barbosa et al., 2021; Moravec et al., 2021), with the period between 2014 and 2018 regarded as the worst multi-year soil moisture drought in Europe during the last 253 years (Moravec et al., 2021). To assess the impact of the drought on carbon cycling in the estuary, we use a longer period between 1997 and 2020, to allow comparisons between non-drought and drought periods. Since 1997, the ecosystem of the Elbe Estuary was designated to be in a recovery state, after heavy pollution in the 1980s, and a transitional state in the 1990s (Rewrie et al., 2023). The authors characterized the current recovery ecosystem state by non-toxic levels of heavy metals, which permitted the reestablishment of autotrophic and heterotrophic processes and OM cycling within the estuary.

The annual mean DIC in the mid to lower Elbe Estuary has been increasing significantly by up to 11 µmol $L^{-1}$ $yr^{-1}$ from 1997 to 2018 (Rewrie et al., 2023), but the source of this increase remains unclear. It was suggested by Rewrie et al. (2023) that an increase in upper estuary TOC over time may provide labile organic carbon available for remineralization in the mid to lower estuary. The organic carbon cycling in the upper Elbe Estuary was evaluated before (Amann et al., 2012), and the study identified that from the late 1990s the POC fraction fueled heterotrophic respiration in the estuary, whereas the removal of DOC was negligible. However, the last decade was not included in this analysis. To address these questions, the current study has extended the recovery state period (Rewrie et al., 2023) by two years with further available data, and aims to (1) identify the reasons for the significant DIC increase, and (2) investigate how the onset of the recent drought has modulated the carbon cycling within the estuary. Data for organic and inorganic carbon content supported by water quality measurements are used to assess the long-term changes in the carbonate system in the Elbe Estuary between 1997 and 2020, with focus on the recent drought.

## 2 Methods and Data

### 2.1 Study site

The temperate Elbe Estuary is a well-mixed, mesotidal coastal plain estuary, with a maximum turbidity zone (MTZ) extending from around Elbe-km 650 to 700 (Fig.1a, Amann et al., 2015). The estuary stretches over 142 km from the tidal border at the Geesthacht Weir to the mouth of the estuary at Cuxhaven, Germany. It connects one of the largest rivers in Northern Europe, the Elbe, to the German Bight in the southern North Sea. The Elbe Estuary was separated into seven

zones designated by the TIDE project (Geerts et al., 2012). In this study, the zones are sub-grouped into five regions: the upper estuary (z1), Hamburg Harbour (z2–z3), middle (mid, z4–z5), lower (z6), and outer (z7) estuary.

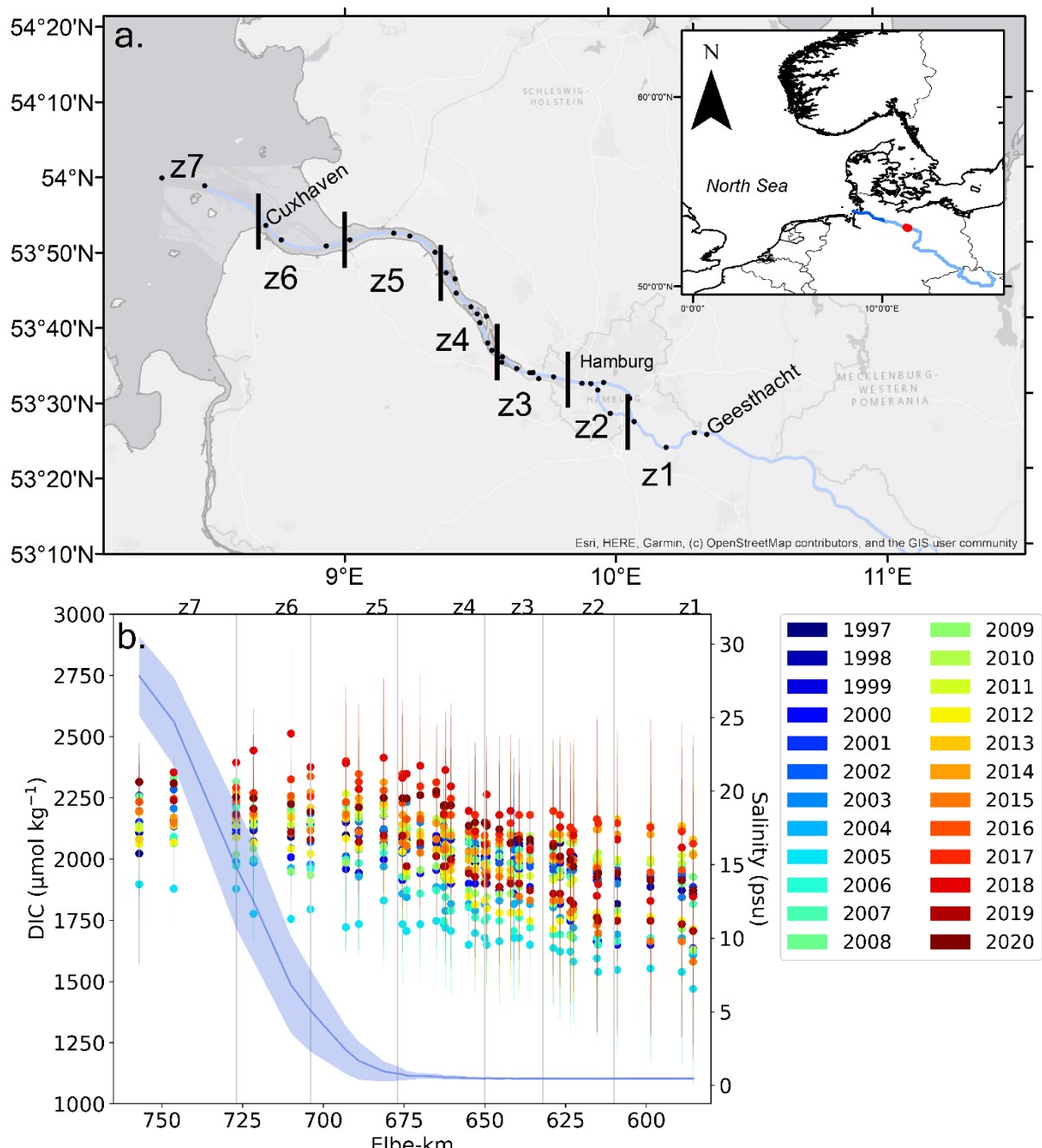

Figure 1: a) Map of the Elbe Estuary, separated into seven zones designated by the TIDE project (Geerts et al., 2012), with sub-groups: the upper estuary (z1), Hamburg Harbour (HH, z2–z3), middle (z4–z5), lower (z6), and outer (z7) estuary. The black dots are the helicopter sampling stations with Geesthacht at 585.5 Elbe-km and Cuxhaven at 727 Elbe-km. The Elbe-km count is the distance from where the Elbe River passes the border between the Czech Republic and

Germany. Insert map: The tidal estuary (dark blue), the non-tidal Elbe River (light blue) and the Neu Darchau gauging station (536.4 Elbe-km, red) are indicated. b) Mean annual DIC in the Elbe Estuary from 1997–2020, with error bars representing the standard deviation from the mean. The mean salinity gradient is shown with dark blue line based on data from 1997 to 2020, with shaded blue area representing one standard deviation around the mean.

## 2.2 Data sources

Data for DIC, POC and key ecosystem parameters (dissolved oxygen (DO), pH, biological oxygen demand ($BOD_7$)) were acquired from the data portal of the Flussgebietsgemeinschaft Elbe (FGG, River Basin Community; https://www.fgg-elbe.de/elbe-datenportal.html) for 1997 to 2020. The FGG Elbe took surface water samples from 36 stations in the estuary (Fig. 1a) by helicopter at full ebb current, which permitted the greatest possible synoptic comparability between the samples with regard to the influence of the tides (ARGE Elbe, 2000). Sampling was generally carried out once per month in February, May, June, July, August and November (see Rewrie et al. (2023) for a detailed description). The samples for DIC were analysed at FGG Elbe laboratories during the analysis of total dissolved carbon, in order to determine both DIC and dissolved organic carbon (DOC). Methods for organic carbon, DIC, dissolved oxygen (DO), pH and $BOD_7$ are listed in Table S1, and further described in Rewrie et al. (2023). The POC was calculated as the difference between measured TOC and DOC (Table S1), with an estimated uncertainty of 20% based on the Pythagorean Theorem (U. Wiegel, pers. comm). The apparent oxygen utilisation (AOU) was calculated:

$$AOU = [O_{2eq}] - [O_2], (1)$$

where *[$O_2$]* is the DO concentration observed and *[$O_{2eq}$]* is the DO concentration expected at equilibrium with air at an absolute pressure of 101325 Pa (sea pressure of 0 dbar) including saturated water vapour. The solubility coefficients derived from Benson and Krause (1984), and as fitted by Garcia and Gorden (1992, 1993) were used.

## 2.3 River discharge

Daily freshwater discharge data (1960–2020) from Neu Darchau gauging station (536.4 Elbe-km, Fig. 1a) were also obtained from the FGG Elbe data portal (https://www.elbe-datenportal.de/). The historical mean river discharge (1960–2020) and the discharge during the

drought period (2014–2020) were calculated, and the significance of the difference between the
two periods (medians) was assessed using the Mann-Whitney-U-test, since both datasets
presented a non-normal distribution from a Shapiro-Wilk test ($p < 0.05$). The significance of a
monthly river discharge trend during the recovery state (1997–2020, Rewrie et al. (2023)) was
assessed with the Pearson Correlation Coefficient.

**2.4 Calculation for carbonate parameters (total alkalinity and $pCO_2$)**

Using the CO2SYS program version 2.5 in Excel (Lewis and Wallace, 1998), the aquatic
carbonate system parameters $CO_2$ partial pressure ($pCO_2$) and total alkalinity (TA) were
calculated from measured DIC, temperature, salinity, pH and, when available, phosphate and
silicate, applying the carbonic acid dissociation constants $K_1$ and $K_2$ of Cai and Wang (1998).
DIC concentrations were reported as whole, rounded numbers in mg L$^{-1}$, and for the carbonate
system calculations, these were converted into µmol kg$^{-1}$. Rewrie et al. (2023) provide an
extensive evaluation of the FGG Elbe DIC data, with an estimated analytical uncertainty of
99.7-102.5 µmol kg$^{-1}$ for DIC. The propagation of uncertainties for the calculated $pCO_2$ and TA
(Orr et al., 2018) were determined using the estimated DIC analytical uncertainty, uncertainties
of the involved constants provided in Orr et al. (2018) and the recommended total standard
uncertainty for pH of 0.01 (Dickson, 1993; Orr et al., 2018).

The median in the Elbe Estuary (z1–z7, 1997–2020) of the CO2SYS program uncertainty
output was 209 µatm for $pCO_2$ (18% uncertainty relative to the mean $pCO_2$) and 100 µmol kg$^{-1}$
for TA (5% uncertainty relative to the mean TA) (Table S2). Calculated TA and $pCO_2$ are
comparable to previously published measured TA values and calculated $pCO_2$ by Amann et al.
(2015) between 2009 and 2011, with similar along-estuary patterns and magnitude. An example
of the comparison for August 2010 is shown in Fig S1. Measured TA and calculated $pCO_2$ in
June 2019 by Norbisrath et al. (2022) are also comparable to the calculated TA and $pCO_2$ in
this study (Table S3).

**2.5 Remineralization of upper estuary POC and DOC**

To assess the remineralization of the upper estuary POC and DOC in the Hamburg Harbour and
the mid-estuary, the percentage decrease in organic carbon ($OC_D$) was calculated according to
the method used by Amann et al. (2012):

$$OC_D\% = \frac{(C_{z1} - C_{zi})}{C_{z1}} \times 100, (2)$$

where C is the mean POC or DOC concentration in the respective zone, with z1 representing zone 1, and with $z_i$ representing zones 2-3 for POC and zones 2-5 for DOC in late spring (May) and summer (June-August) from 1997 to 2020. The POC decrease, compared to zone 1, was only calculated for zone 2 and 3 due to the influence of the maximum turbidity zone in zone 4

and 5 (Amann et al. 2012). Negative values indicate OC addition.

**2.6 Statistical analyses for along-estuary carbonate parameters and upper estuary POC**

For every zone, the Pearson Correlation Coefficient was applied to the winter (February),

autumn (November), late spring (May) and summer (June, July and August (JJA)) DIC, TA and $pCO_2$ records to assess the trend and seasonal change from 1997 to 2020. The Pearson Correlation Coefficient was also applied to the late spring and summer POC in the upper region (z1) over time, when we assume that z1 represents the river input into the estuary. The difference in DIC concentration ($\Delta$[DIC]) between the mid, lower and outer estuary (z4–z7,

represented by zi) relative to the upper region (z1) was used to determine the along-estuary DIC gain over time (1997–2020) in late spring and summer:

$$\Delta[DIC] = [DIC]_{(zi)} - [DIC]_{(z1)}, (3)$$

The Pearson Correlation Coefficient was used to assess the dependency of along-estuary DIC gain ($\Delta$DIC) on upper estuary POC concentration (z1).

**2.7 Upper estuary POC loads as a driver of DIC loading**

POC loads in the upper estuary (z1) and DIC loads in the estuary in Gmol C month[-1] were calculated by multiplying discharge (Q) and carbon (DIC or POC) concentration:

$$L = Q \times [C], (4)$$

Monthly mean POC and DIC for each zone in May to August were calculated from 1997 to 2020, $Q$ was the respective mean monthly river discharge ($m^3 s^{-1}$) recorded at the Neu Darchau gauging station. A correction factor to the monthly river discharge was applied to each estuarine zone (zones 1 to 6) to account for tributary inputs along the estuary (Amann et al., 2015). Error

bars indicate the standard deviation ($\sigma$) of the product:

$$\sigma_{(L)} = \sqrt{(\frac{\sigma_{[C]}}{[C]})^2 \times (\frac{\sigma_Q}{Q})^2} \times L \, , (5)$$

The difference in DIC loads ($\Delta DIC_{(L)}$) between the upper region (z1) and the mid and lower region (z4-z6, represented by zi) were quantified for each month to estimate the internal DIC load:

$$\Delta DIC_{(L)} = L_{(DIC,zi)} - L_{(DIC,z1)} \, , (6)$$

Negative values indicate DIC loss within the estuary, while positive values indicate DIC gain. Error bars indicate the standard deviations ($\sigma$) of the difference:

$$\sigma_{\Delta DIC_{(L)}} = \sqrt{\sigma_{L(zi)}^2 + \sigma_{L\,(z1)}^2} \, , (7)$$

The statistical differences between internal DIC loads during the recent drought (2014–2020) and non-drought (1997–2013) periods, and the differences between the internal DIC load in the mid-lower estuary and the POC load in the upper estuary (z1), for May to August were tested. The independent *t*-test was used for datasets that presented a normal distribution from a Shapiro-Wilk test (p < 0.05), and the Mann-Whitney U Test was applied to the datasets that presented a non-normal distribution. The months of August 2002, August 2010 and June–July, 2013 were excluded from the statistical analysis as anomaly flood months (Kienzler et al., 2015; Voynova et al. 2017).

## 2.8 Water-air CO$_2$ exchange

To estimate the inorganic carbon export dynamics in the estuary, the flux of CO$_2$ between water and atmosphere in mol m$^{-2}$ d$^{-1}$ was estimated for each sampled station, in the upper to lower region (Fig. 1a), between 1997 and 2020 with the equation:

$$F = k \times \alpha \times (pCO_{2\,(water)} - pCO_{2\,(atmosphere)}), (8)$$

where *k* is the gas transfer velocity, and $\alpha$ is the solubility coefficient of CO$_2$ (calculated from Weiss 1974: in mol L$^{-1}$ atm$^{-1}$). $pCO_{2(water)}$ was calculated from the FGG Elbe DIC and pH samples, and $pCO_{2(atmosphere)}$ was calculated according to Dickson et al. (2007) with:

$$pCO_{2\,(atmosphere)} = XCO_2 \times (P_{ATM} - pH_2O), (9)$$

where $XCO_2$ is the molar fraction of CO$_2$ in dry air obtained from the Global Monitoring Laboratory (Lan et al., 2023). Daily mean ambient air pressure ($P_{ATM}$) from E-OBS meteorological data for Europe (Cornes et al., 2018) from the Copernicus Climate Data Store (https://cds.climate.copernicus.eu) was selected for the Elbe Estuary region and each sampling

station (Fig. 1). The saturated water partial pressure $pH_2O$ in atm was derived according to Weiss and Price (1980) with:

$$\ln pH_2 O = 24.4543 - 67.4509 \left(\frac{100}{T}\right) - 4.8489 \ln\left(\frac{T}{100}\right) - 0.000544\ Sal, (10)$$

where T denotes in-situ temperature in Kelvin and Sal is the salinity of the sample. The gas transfer velocity $k$ was calculated after Wanninkhof (2014) as follows:

$$k = 0.251 \times (U_{10})^2 \times \left(\frac{Sc}{600}\right)^{-0.5}, (11)$$

where 0.251 is the coefficient of gas transfer, $U_{10}$ is the wind speed in m s$^{-1}$ measured *in situ* at 10 m height from E-OBS meteorological data for Europe (Cornes et al., 2018 from https://cds.climate.copernicus.eu). The Schmidt number (*Sc*) for $CO_2$ in freshwater was calculated according to Wanninkhof (2014) as a function of water temperature. The uncertainty estimate of $k$ has been estimated to 20 % (Wanninkhof, 2014). We find a good fit of water-air $CO_2$ flux estimates to those calculated in Norbisrath et al. (2022), as shown in Table S4. We used the total area of the Elbe Estuary of 276.6 km$^2$ between Geesthacht and Cuxhaven, Germany, originally derived via GIS by Amann et al. (2015), to estimate the water-air estuary $CO_2$ flux in Gmol C yr$^{-1}$.

**3 Results**

Two main features are notable in the mean annual DIC concentration: (1) DIC increased from the upper freshwater estuary towards the mid to lower estuary (Fig. 1b), suggesting along-estuary accumulation of DIC, with a maximum in the MTZ and lower estuary (z5–z6); (2) a pronounced DIC increase over time, reaching a maximum mean annual DIC in the lower estuary (z6, 2512 ± 349 µmol kg$^{-1}$) in 2018, which was 20% higher than in 1997 (2090 ± 364 µmol kg$^{-1}$). This is a distinctive feature of the along-estuary DIC pattern for the recovery ecosystem state (Rewrie et al. 2023).

**3.1 Drivers of DIC dynamics along the estuary and DIC increase in the Elbe Estuary**

Since 1997, the lowest DIC in the upper region in late spring and summer coincided with high pH (9.4) and large negative AOU (-288 µmol L$^{-1}$), i.e. oxygen supersaturation with respect to atmospheric equilibrium. This suggests that dominating autotrophy depletes DIC in the upper estuary, and most likely the upstream river regions, which is supported by highest seasonal

chlorophyll *a* concentrations in May to August at 585.5 Elbe-km, reaching $166 \pm 74\,\mu g\,L^{-1}$ (Fig. S2). The exception was between 2018 and 2020, when pH decreased to 7.7, and AOU had predominately positive values (Fig. 2d), up to $+117\,\mu mol\,L^{-1}$, indicating a possible shift to dominating heterotrophy in z1, 4–6 years after the onset of the drought in 2014 (Fig. 2a).

The different estuarine regions (Fig. 1a) are clearly distinguishable in pH (Fig. 2c) and AOU (Fig. 2d). Between the Hamburg Harbour and the lower estuary in late spring to summer, pH decreased compared to the upper region, and AOU was positive, coupled with an along-estuary increase in DIC. This suggests dominating heterotrophic activity during the warmer months (see also Amann et al., 2015; Rewrie et al., 2023) and accumulation of DIC in surface waters. In the outer estuary, pH increased and AOU was predominately negative, indicating dominating autotrophy in the coastal regions adjacent to the estuary. Changes in the along-estuary DIC concentrations over time (1997–2020) appear to be decoupled from the pH and AOU dynamics (Fig. 2b–d). When spring and summer DIC concentrations were lowest in 2005–2006, ranging between $914\,\mu mol\,kg^{-1}$ and $2040\,\mu mol\,kg^{-1}$, this minimum was not reflected in concurrent change in AOU or pH.

Compared to the upper estuary (z1), the late spring and summer DOC and POC decreased on average by $0.3 \pm 21\%$ and $40.6 \pm 18\%$, respectively, in Hamburg Harbour (z2-z3) and mid-estuary (only DOC in z4-z5), for the period 1997–2020 (Fig. 3). This corresponded to a mean concentration decrease of $10 \pm 69\,\mu mol\,kg^{-1}$ and $157 \pm 106\,\mu mol\,kg^{-1}$ in DOC and POC. This indicates that respiration of upper estuary POC, rather than DOC, dominates heterotrophic activity in the Hamburg Harbour and mid Elbe Estuary, and subsequent DIC production.

Significant mean POC increases occurred in late spring (May, $14\,\mu mol\,C\,kg^{-1}\,yr^{-1}$) and summer (June–August, $8\,\mu mol\,C\,kg^{-1}\,yr^{-1}$) in the upper estuary (Fig. 4, Table 1) from 1997 to 2020. A significant concurrent decrease of $BOD_7$ in late spring and summer (Fig. S3), suggests an improvement in water quality. This indicates a long-term intensification in production of OM in the upper estuary (z1) and in the river upstream of this region. Four years after the onset of the drought, POC dropped by 35% in summer 2018–2020 ($325 \pm 141\,\mu mol\,kg^{-1}$), and this lower POC coincided with the shift to lower pH (Fig. 2c), and predominately positive AOU (Fig. 2d).

Coincident with the 1997–2020 POC increase in the upper estuary, DIC and TA increased significantly in the mid to outer estuary by up to $21\,\mu mol\,kg^{-1}\,yr^{-1}$ in late spring (May) and 12

$\mu$mol kg$^{-1}$ yr$^{-1}$ in summer (June–August), with a significant positive correlation between mid-estuary DIC gain and upper estuary POC content in late spring. Also, in late spring, DIC concentration peaked in the mid-estuary (z5, 1997–2019) during 73% of the measurements over the past 10 years (2009–2019), indicating a stabilization in the estuarine DIC cycling pattern. In both late spring and summer, the DIC gain in the mid-estuary ($286 \pm 247$ to $359 \pm 155$ $\mu$mol kg$^{-1}$) was not significantly different from the upper estuary POC ($347 \pm 94$ to $377 \pm 165$ $\mu$mol kg$^{-1}$, Table S5 & S6), indicating that, the amount of upstream POC (z1) available for remineralization was sufficient to account for the production of mid-estuary DIC. In addition to this evidence, the ratio of TA to DIC can serve as an indicator of the source of carbon, and specifically when < 1 this can reflect DIC input in the form of $CO_2$ (Joesoef et al., 2017), which was observed in this temperate estuary. The mid-estuary was characterized by a TA:DIC ratio < 1.0 (z4, Fig. 4), with the highest $p$CO$_2$ content, exceeding > 1000 $\mu$atm in late spring and summer, indicating that the additional DIC input was in the form of $p$CO$_2$.

During summer, in 73% of all years, the along-estuary DIC was highest in the outer estuary between 1997 and 2020. This differs from the mid-estuary DIC peak in late spring (May) and suggests production in, or lateral transport of DIC into the outer estuary in summer. The DIC variability along the salinity gradient, with conservative mixing line between the river and North Sea end members, showed non-conservative behaviour, i.e. positive (43%) and negative (23%) deviations from the mixing line in May to August (Figs. S4-S7). The positive deviations indicate mainly an internal source of DIC in the lower to outer estuary. Mean summer AOU in the outer estuary was negative in 61% of all years (1997–2020). During these years, the mean DIC was significantly and positively correlated with mean AOU ($r = 0.58$, $p < 0.05$) and negatively correlated with mean pH ($r = -0.58$, $p < 0.05$). These correlations demonstrate a control of primary producers on DIC in the outer estuary during summer. When AOU was > 0, there was no trend in DIC with AOU and pH. Along the estuary, the TA to DIC ratio increased to > 1.0 (Fig. 4), while $p$CO$_2$ decreased to values ranging from $65 \pm 58$ $\mu$atm to $821 \pm 363$ $\mu$atm, in the outer estuary, suggesting drawdown of $CO_2$ in this region.

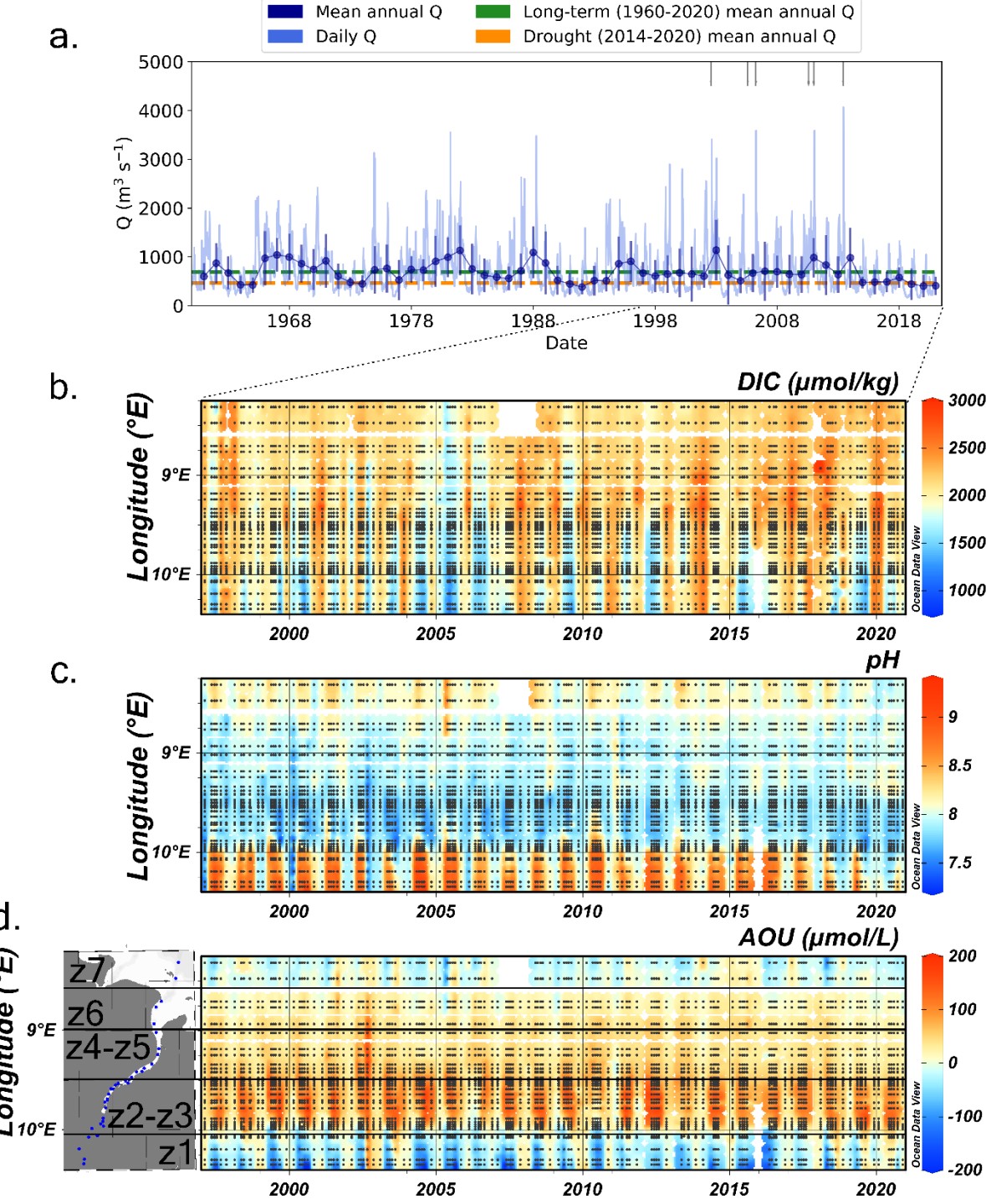

Figure 2. a) Daily Elbe River discharge (light blue), mean annual river discharge with error bars representing the standard deviation of the mean (dark blue) and flood events (grey marks) between 1960 and 2020. Long-term (1960–2020) mean annual Elbe River discharge of 690 ± 441 m$^3$ s$^{-1}$ (green dashed line) and the drought period (2014–2020) mean annual Elbe River discharge of 468 ± 234 m$^3$ s$^{-1}$ (orange dashed line). Hovmöller diagram of (b) DIC (µmol kg$^{-1}$), (c) pH and (d) apparent oxygen utilization (AOU in µmol L$^{-1}$), with map of sampling stations (also refer to Fig. 1a) and black lines separating the upper (z1), Hamburg Harbour (z2–z3), mid

(z4–z5), lower (z6), and outer (z7) regions, in the Elbe Estuary from 1997–2020 (note different time-scale to the river discharge). The Hovmöller diagrams were produced with DIVA gridding in Ocean Data View and black dots represent the sampling stations.

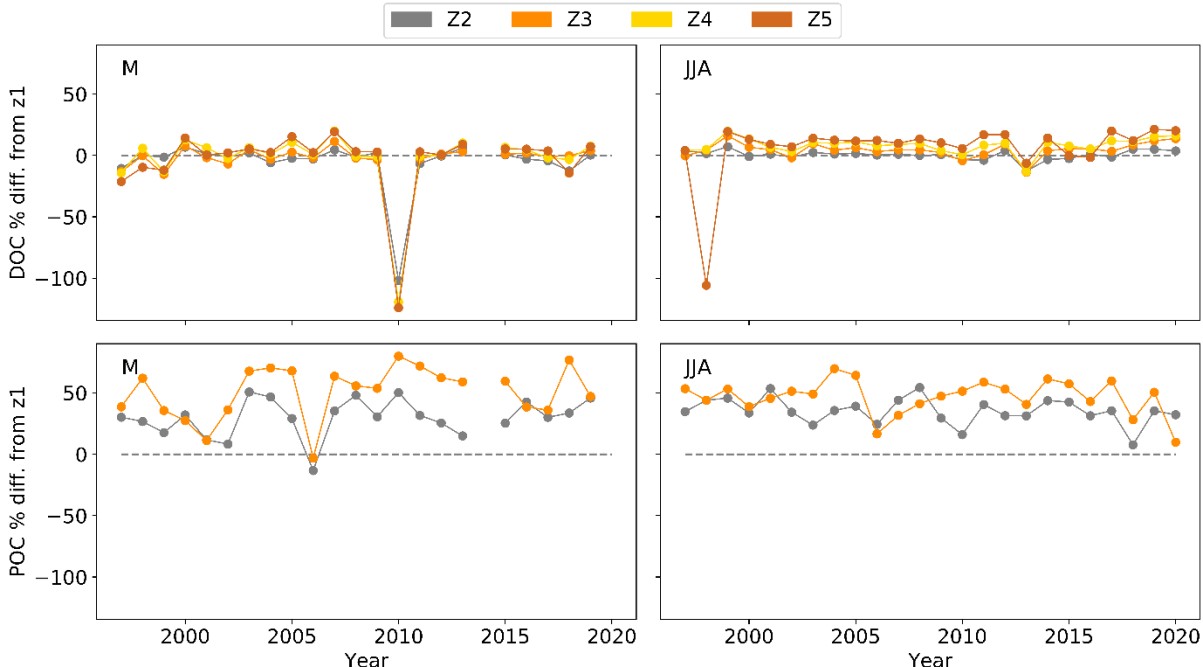

Figure 3. The OC percent (%) difference (diff.) in zones 2-5 (DOC) and zones 2-3 (POC) from initial OC in the upper estuary (zone 1 (z1)). The percent decrease was calculated based on Eq. 2 to estimate the OC removal in zones 2-5 for DOC and zones 2-3 for POC, compared to the upper estuary OC (z1), in late spring (May, M) and summer (June-August, JJA) from 1997 to 2020, with indicated zero removal (dashed grey line).

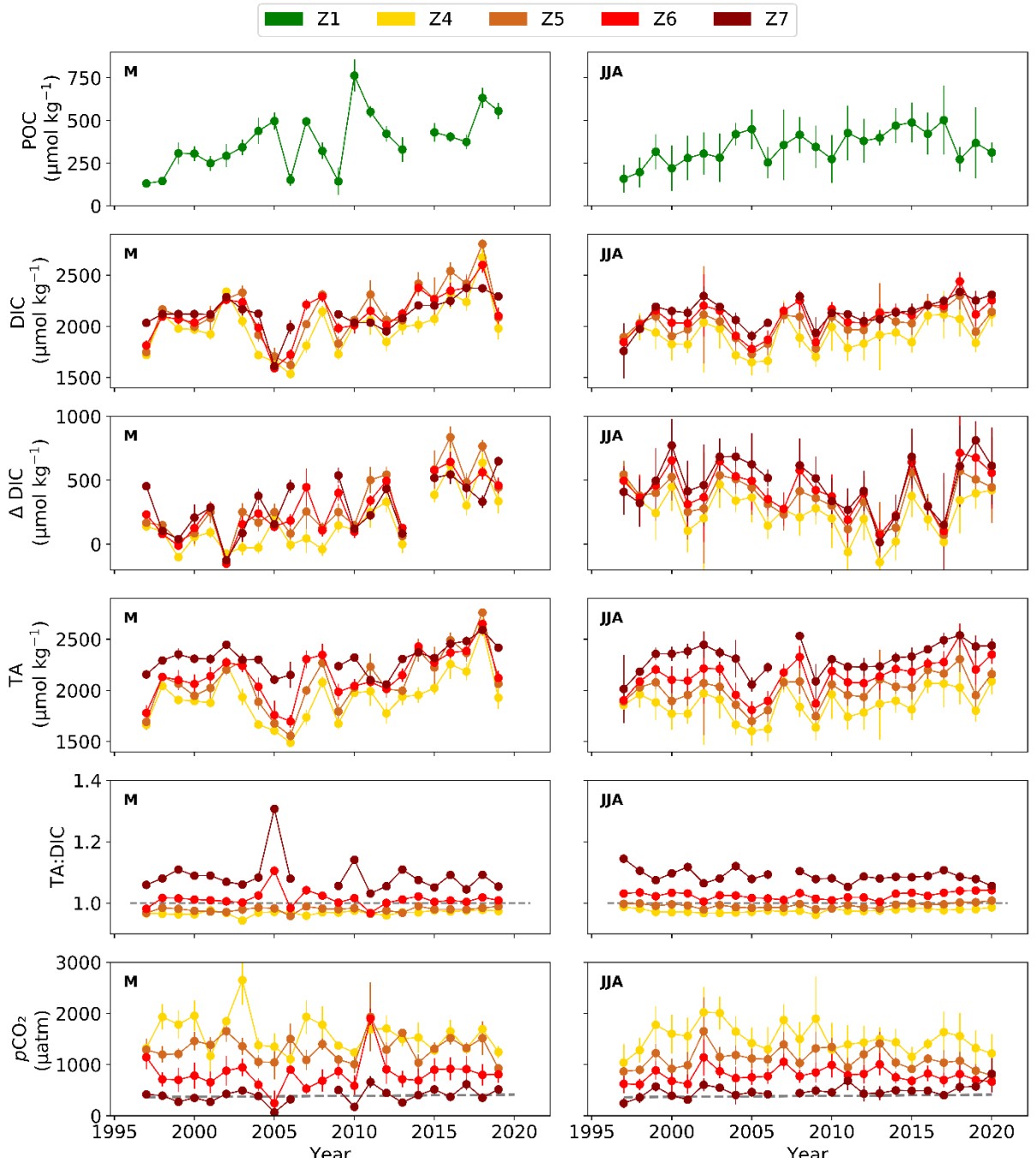

Figure 4. Late spring (May, M) and summer (June–August, JJA) POC in zone 1 (Z1), and in the mid to outer estuary (zone (Z) 4–7, Fig. 1b) average DIC, along-estuary DIC gain (ΔDIC), TA, TA:DIC ratio (1 indicated by grey dashed line), and $p$CO$_2$, with atmospheric CO$_2$ values (grey dashed line) from Global Monitoring Laboratory (Lan et al., 2023) from 1997 to 2020 (in May to 2019). Note differences in y-axis scales. Error bars represent the standard deviations of the mean.

Table 1. Pearson Correlation Coefficient (*r*) of late spring (May) and summer (June–August (JJA)) POC concentration in zone 1 ([C]), and in zones 4–7 average DIC concentration ([C]), the along-estuary DIC gain (ΔDIC) and TA concentration in the Elbe Estuary with time (decimal year). Highlighted are two levels of significance: $p < 0.05$ (*) and $p < 0.10$ (#). The rate of change is given for POC, DIC and TA (β, $\mu mol\ kg^{-1}\ yr^{-1}$), with the respective standard error (se). Pearson Correlation Coefficient was also given between the along-estuary DIC gain (ΔDIC) with POC concentration in zone 1 (z1). The Shapiro-Wilk test of normality was applied prior to statistical analysis. The Spearman Rank Correlation was applied to DIC in zone 7, ΔDIC in zone 4 and ΔDIC in zone 4 with POC concentration in zone 1 for May. Data for May from 1997–2019.

| | Month | Zone | [C] vs time (1997–2020) | | ΔDIC vs time (1997–2020) | ΔDIC Vs POC (z1) | [TA] vs time (1997–2020) | |
|---|---|---|---|---|---|---|---|---|
| | | | r | β (se) ($\mu mol\ kg^{-1}\ yr^{-1}$) | | | r | β (se) ($\mu mol\ kg^{-1}\ yr^{-1}$) |
| POC | May | 1 | 0.58* | 14 (4) | | | | |
| | JJA | 1 | 0.57* | 8 (2) | | | | |
| DIC, ΔDIC or TA | May | 4 | 0.40# | 15 (7) | 0.67* | 0.44* | 0.42* | 16 (8) |
| | | 5 | 0.51* | 21 (8) | 0.73* | 0.43* | 0.52* | 21 (7) |
| | | 6 | 0.50* | 17 (6) | 0.71* | 0.30 | 0.49* | 16 (6) |
| | | 7 | 0.44* | 9 (5) | 0.53* | 0.00 | 0.37# | 7 (4) |
| | JJA | 4 | 0.29 | 6 (4) | -0.26 | -0.47* | 0.31 | 7 (4) |
| | | 5 | 0.42* | 8 (4) | -0.19 | -0.39# | 0.44* | 9 (4) |
| | | 6 | 0.50* | 11 (4) | -0.06 | -0.30 | 0.51* | 12 (4) |
| | | 7 | 0.50* | 10 (4) | -0.10 | -0.19 | 0.42* | 8 (4) |

**3.2 Influence of drought on estuarine DIC**

From 2014 to 2020, the mean annual Elbe River discharge of $468 \pm 234$ $m^3\ s^{-1}$ was 32% lower than the long-term 1960–2020 mean at $690 \pm 441$ $m^3\ s^{-1}$ (Fig. 2a, Table S8). This confirms that an inter-annual hydrological drought took place since 2014, characterized by overall reduced streamflow in the Elbe River (Zink et al., 2016). Between 1997 and 2020, May was the only month with a significant negative trend in mean monthly river discharge ($r = -0.43$, $p < 0.05$), reaching lowest discharge of $264 \pm 19$ $m^3\ s^{-1}$ in May 2020. Such low monthly discharge is usually observed during dry summer and early autumn months, with more extreme values in 2018–2019 at $< 200$ $m^3\ s^{-1}$. This suggests that the drought extended the low discharge summer period into late spring, most likely increasing the water residence time in the estuary in late spring, as seen previously by Bergemann et al. (1996).

Despite this significant decrease in river discharge, in May, the internal DIC load in the mid to lower estuary was significantly higher during the drought period of 2014 to 2020, at $0.9 \pm 0.22$ Gmol C month$^{-1}$, compared to the non-drought period of 1997 to 2013 at $0.5 \pm 0.27$ Gmol C month$^{-1}$ (Fig. 5, Table S9–S10). In summer, the internal DIC load decreased significantly, by 35–42%, during the recent drought years (2014–2020), down to $0.3 \pm 0.18$ Gmol C month$^{-1}$

(June z4 and July z5–z6, Table S9–S10), compared to the non-drought period, ranging between $0.5 \pm 0.27$ and $0.7 \pm 0.32$ Gmol C month$^{-1}$ (1997–2013). The positive AOU and lower pH in 2018–2020 (Fig. 2c–d), suggest there was a shift from net autotrophy to net heterotrophy in the upper estuary, potentially leading to DIC production and larger input in the upper estuary. This would explain the elevated DIC concentrations in July–August 2017 for example (Fig. S8).

That is, the dominating heterotrophy in recent years (2018–2020) and subsequent DIC generation in the upper region (z1) could reduce the DIC load difference between the upper and mid estuary. In turn, this reduces the internal DIC load in the mid estuary (Fig. 5).

Overall, from May to August, the mid to lower estuary internal DIC load (z4–z6, Table S9–

S10) during the drought ($0.5 \pm 0.33$ Gmol C month$^{-1}$) was not significantly different from the non-drought period ($0.6 \pm 0.31$ Gmol C month$^{-1}$). Therefore, during the drought, the May increase in internal DIC load was countered by an observed decrease in summer, ultimately resulting in no net change in the internal DIC load compared to the non-drought period (1997–2013). Albeit, the significant decrease in annual mean river discharge during the drought does

significantly impact seasonal DIC production. Therefore in estuaries such as the Elbe Estuary, it is imperative to consider seasonality in carbon budget calculations.

In late spring (May) and summer (June–August), the mean POC load ($0.3 \pm 0.16$ to $0.6 \pm 0.29$ Gmol C month$^{-1}$) in the upper estuary had the same magnitude as the internal DIC load ($0.3 \pm$

$0.25$ to $0.7 \pm 0.28$ Gmol C month$^{-1}$) in the mid-estuary (1997–2020, Table S12), with no significant difference (z5 in May and z4 in June–August, Table S13). In contrast, the internal DIC load ($0.5 \pm 0.26$ to $0.8 \pm 0.30$ Gmol C month$^{-1}$) in the lower estuary was significantly (1.3-1.9 times) larger than the upper estuary POC load (Table S12–S13). This means that while POC from the upstream regions can account for the DIC production in the mid-estuary, in the lower

estuary, an additional source of OM likely contributes to the DIC production therein.

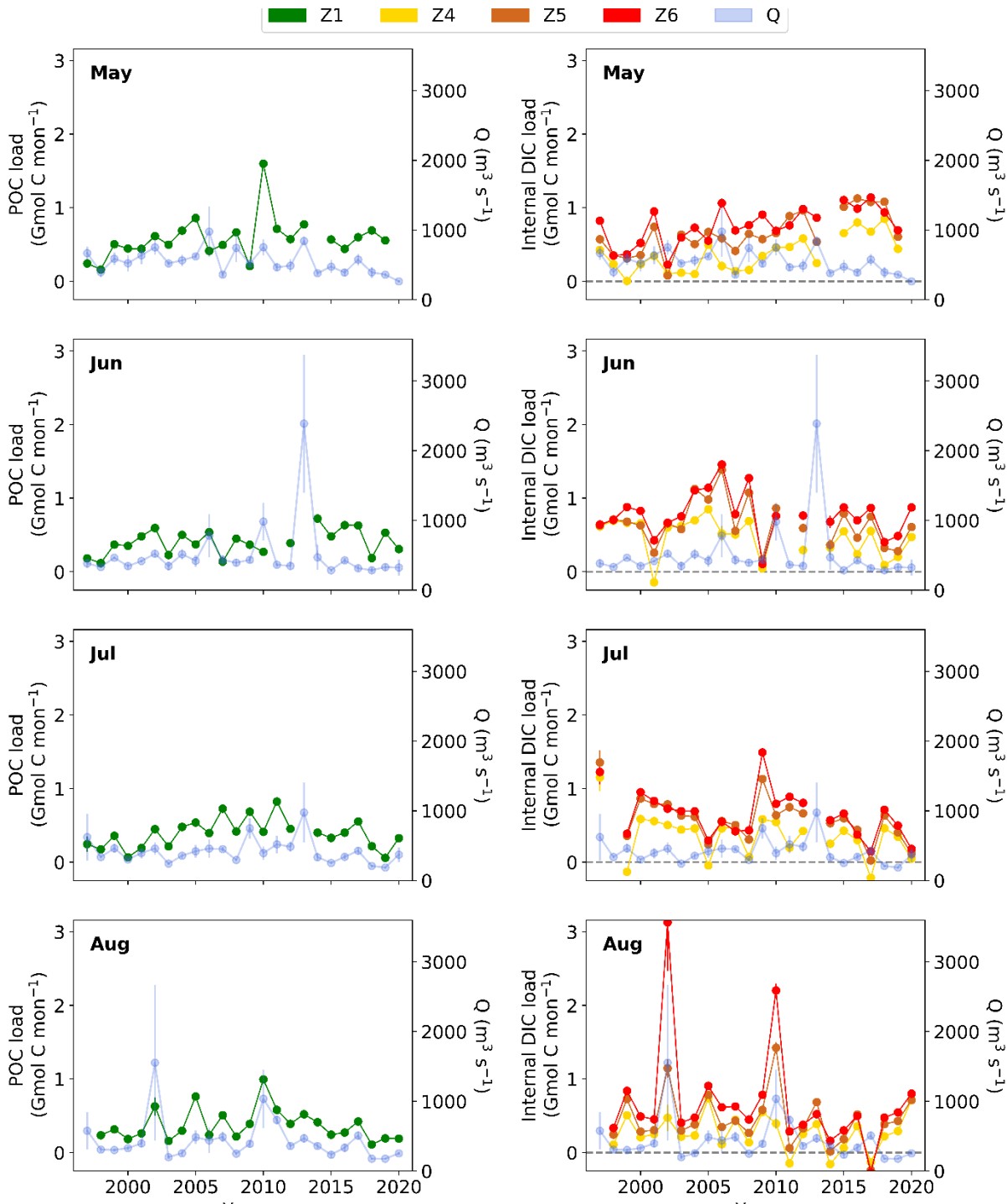

Figure 5. Carbon load (Gmol C month$^{-1}$) as POC in zone 1 (Z1) and the internal DIC load in the mid–lower estuary (zone (Z) 4–6), with indicated zero (dashed grey line), and the respective monthly mean river discharge (Q) for each year (light blue). Error bars represent the standard deviations of the mean.

### 3.3 Annual inorganic carbon export estimates

The annual inorganic carbon export to the atmosphere and adjacent coastal waters was estimated to evaluate the overall inorganic carbon export dynamics in the Elbe Estuary between 1997 and 2020. The average annual water-to-air $CO_2$ flux was $6 \pm 1.6$ Gmol C yr$^{-1}$ ($21 \pm 5.8$ mol C m$^{-2}$ yr$^{-1}$), with a range between 4 and 10 Gmol C yr$^{-1}$. The highest flux was recorded in 2020 (Fig. 6). The DIC export to adjacent coastal waters, based on the annual lower estuary DIC load (Eqs. 4–5), ranged from $32 \pm 0.9$ to $89 \pm 4.8$ Gmol C yr$^{-1}$.

During the drought period (2014–2020), the DIC export to coastal waters ($38 \pm 5.4$ Gmol C yr$^{-1}$) was significantly lower, on average by 24–31%, compared to the non-drought period ($50 \pm 6.4$ Gmol C yr$^{-1}$ (excl. flood years) and $55 \pm 14.0$ Gmol C yr$^{-1}$ [1] (incl. flood years), Table S15). In contrast, the annual water-to-air $CO_2$ flux was not significantly different during the drought ($6 \pm 1.9$ Gmol C yr$^{-1}$) and non-drought period ($6 \pm 1.6$ Gmol C yr$^{-1}$ (excl. flood years) and $6 \pm 1.5$ Gmol C yr$^{-1}$ [1] (incl. flood years), Table S15).

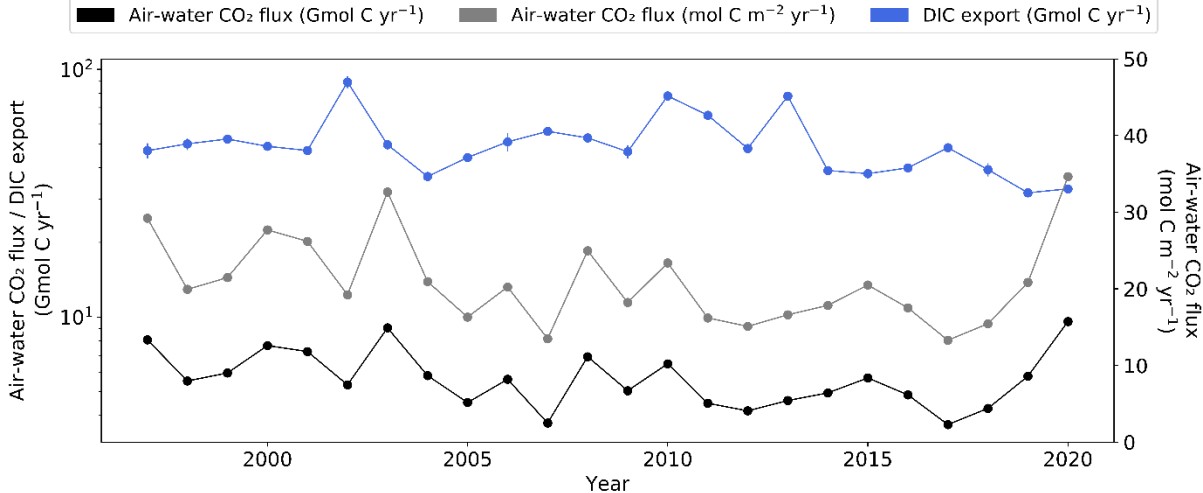

Figure 6. Annual mean water-air $CO_2$ flux estimates in the Elbe Estuary and the annual mean DIC export, based on the lower estuary DIC load (Eqs. 4–5), in Elbe Estuary. Note differences in y-axis scales. Error bars represent the standard deviations of the mean.

### 4 Discussion

Since 1997, the upper estuary and upstream regions have experienced a significant increase in POC in late spring and summer. Coupled with elevated pH reaching 9.4 and negative AOU,

this is evidence that dominating autotrophy in these upstream regions has become a larger source of labile POC to the mid–lower estuary between 1997 and 2020. Kamjunke et al. (2021) also reported high POC (> 749 µmol L$^{-1}$) in the lower Elbe River during summer 2018, with a
strong correlation between POC and chlorophyll-a (chl-a), indicating a dominant contribution of phytoplankton to POC upstream of the Elbe Estuary. We suggest that the underlying reason for this significant increase in phytoplankton production is due to the amelioration of water quality in the river and upper estuary (IKSE 2010; Langhammer 2010; IKSE 2018), indicated by a BOD$_7$ decrease by more than half in summer (1997–2020, Fig. S3). The summer mean
BOD$_7$ decreased from 12 ± 1.7 mg L$^{-1}$ in 1997–2005 to 8 ± 1.1 mg L$^{-1}$ in 2006–2020. While there was a continuous decrease in nutrients from the late 1990s (Wachholz et al. 2022), the nutrient supply was still sufficient to support phytoplankton production (Kamjunke et al., 2021; Dähnke et al 2022). This is evidence that the ecosystem state of the Elbe Estuary, and perhaps the Elbe River, is still changing during the recovery state, as suggested by Rewrie et al. (2023),
following the major anthropogenic influences and social changes before and during the 1980s–1990s. Therefore, changes in water quality should be taken into account in regional and global estimates for carbon processing in estuaries. We also found that the along-estuary DIC concentrations were a function of the DIC source concentrations in the upper estuary, likely due to a DIC drawdown by primary producers in the upstream regions in the Elbe River (e.g.
in 2005–2006 Fig. 2). This highlights the influence of the upper regions as a source of carbon to the estuary, and the importance to evaluate the carbon dynamics from the watershed as well.

In late spring, the significant correlation between mid-estuary DIC gain and upper estuary POC (Table 1) suggests that the increase in OM in the upper estuary and upstream waters is driving
the long-term DIC increase in the mid-estuary. The upper estuary POC in late spring and summer tripled since the onset of the recovery state in 1997, which we suggest is driven largely by allochtonous POC produced in the Elbe River and autochthonous POC produced in the upper estuary. Abril et al. (2002) reported that POC mineralisation efficiency (i.e., the percentage of POC mineralized) is a linear function of POC concentration, and considering the increased POC
concentration, we can expect a higher turnover of POC in the estuary in recent years. That is, from 1997 to 2020, the increase in POC in the upper estuary enhanced the availability of POC for remineralization in the Elbe Estuary, and subsequently increased DIC production, as evidenced by the concomitant increase in DIC estuarine concentration (Fig. 4).


Moreover, the magnitude of DIC gain in the mid-estuary and POC input into the estuary show no significant difference in late spring and summer (Table S5). There was on average 40.6 ± 18% decrease of z1 POC, compared to only 0.3 ± 21% decrease of z1 DOC in the estuary (Fig. 3), suggesting that heterotrophic respiration and DIC production was mainly fuelled by

remineralization of POC, which is in agreement with the findings of Amann et al. (2012). There are also no other major sources of carbon along the estuary (Abril et al., 2002), suggesting that POC was efficiently remineralized and converted to DIC by the mid-estuary. This is supported by Abril et al. (2002) who reported mineralisation of labile riverine POC simultaneously with an increase in SPM at the entrance of the MTZ (here z4). We find that POC drops to < 4% of

SPM in May to August (1997–2020) in the mid estuary (z4–z5, Fig. S9), indicating widespread OM remineralization in the estuary. This suggests that improving water quality in an ecosystem, following significant ecosystem state changes (Rewrie et al., 2023), can result in an increase in the estuary's efficiency as a natural filter.

A large part of POC from the Elbe is mineralized in the oxygen minimum zone in Hamburg Harbour (z2–z3), characterized by large removal of POC (Fig. 3), and by persistent low pH and positive AOU (Fig. 2c–d). This was also previously suggested by Amann et al. (2012). A low carbon to nitrogen ratio of suspended matter, with high dissolved silicate concentration, found by Dähnke et al. (2022) corroborates this, most likely with the dissolved silica values coming

from diatom frustules. As a result of the POC remineralization, $pCO_2$ increases exponentially from Hamburg Harbour (e.g. Fig S1), and reaches highest $pCO_2$ in the mid-estuary (z4), between 2.9 and 7.1 times that of the global annual mean atmospheric $CO_2$ (Fig. 4). Similar $pCO_2$ ranges were previously calculated in the Elbe Estuary (Amann et al., 2015; Norbisrath et al., 2022), and the calculated $pCO_2$ values here match data from August 2010 (Fig. S1) and

June 2019 (Table S3). We postulate that POC from the upper estuary is therefore efficiently remineralized in the Hamburg Harbour, and is converted into DIC as $CO_2$, in the low pH regions (z2–z3, Fig. 2) and mid-estuary (z4–z5), also shown by Amann et al. (2015). As along-estuary pH increases in the mid to the lower estuary (Fig. 2c), accompanied by an along-estuary decrease in $pCO_2$ (Fig. 4), a speciation shift from $pCO_2$ to bicarbonate ($HCO_3^-$) occurs, retaining

$CO_2$ in the carbonate system buffer. Previous studies (Kempe, 1982; Brasse et al., 2002) reported that seawater enriched in carbonate ($CO_3^{2-}$) buffers by titration the freshwaters, high in $pCO_2$ and undersaturated with respect to calcite, in the low-salinity (< 15 psu) mid to lower estuary regions, thus preventing $CO_2$ outflux to the atmosphere. An upper bound of 90% TA generation via calcium carbonate dissolution as described by Norbisrath et al. (2022) within the

Elbe Estuary can also convert the high $CO_2$ to $HCO_3^-$. These could explain the observed continual along-estuary gain of DIC. Internal remineralization of POC via respiration in the Elbe Estuary, especially by the mid-estuary, and the equivalent magnitude of POC concentration and along-estuary DIC gain, indicate retention of carbon in the Elbe Estuary, rather than a significant loss to the atmosphere. The mid and lower estuary concurrent increase in TA over time (Table 1 and Fig. 4), and absence of temporal $pCO_2$ increase (Fig. 4), indicates that the produced DIC in the form of $pCO_2$ was consistently converted to $HCO_3^-$. These findings emphasize the significance of the estuary in the carbon processing along the land-to-ocean continuum, when an estuary is not affected by major pollution (Rewrie et al. 2023).

Increasing alkalinity trend during the past century has been observed in river and estuarine systems (Raymond et al., 2008; Kaushal et al. 2013). However, we propose that the temporal late spring and summer DIC increase in the Elbe Estuary was not fuelled by internal production of TA nor by external inputs of TA via the Elbe River. Norbisrath et al. (2022) suggested that calcium carbonate dissolution is a main biogeochemical process producing TA in the Elbe Estuary, which increases TA and DIC at a 2:1 ratio (Guo et al., 2008), and would therefore result in an overall higher TA concentration compared to DIC. The higher DIC content compared to TA in the mid-estuary (Fig. 4) suggests instead that the along-estuary increase in DIC late spring and summer was not fuelled by a change in $CaCO_3$ dissolution processes. The rise of TA in rivers can increase TA in coastal ecosystems, as shown by a nearly 50% increase in TA export from the Mississippi River to the Gulf of Mexico (Raymond et al., 2008), which was found to be anthropogenically driven via cropland expansion, coupled with increased precipitation in the river catchment. In the present study, the absence of late spring and summer TA increase in the upper and Hamburg Harbour regions (z1–z3, not shown) over time (1997 – 2020), suggests there was no apparent change in the TA content of the river waters entering the estuary. However, changes in the carbonate parameters in the Elbe River catchment area should be further investigated.

The calculated TA measurements could be influenced by the variability of OM (Kim et al., 2006; Kuliński et al., 2014). The CO2SYS program does not account for contribution of organic alkalinity in the calculated TA. While organic alkalinity typically constitutes a smaller fraction of TA compared to that provided by the inorganic compounds, it could still be a significant component of TA in systems influenced by dissolved OM inputs (Kuliński et al., 2014). For instance, Hunt et al. (2011) reported a 21-100% contribution of organic alkalinity to TA in 15

rivers of northern New England (USA) and New Brunswick (Canada), with low TA (116 µM to 956 µM) and extremely high DOC, up to 1480 µmol L$^{-1}$. In the Vistula and Oder Rivers, characterized by an average TA and DOC concentrations of 2965 ± 568 µmol kg$^{-1}$ and 560 ± 77 µmol kg$^{-1}$, organic TA contributed < 8% (Kuliński et al., 2014). Kuliński et al. (2014) reported that the higher percentage of organic alkalinity contributing to TA in the rivers investigated by Hunt et al. (2011) was due to the lower amount of TA. Mean DOC and TA in the Elbe Estuary (z1-z7) were respectively 498 ± 92 µmol kg$^{-1}$ and 1985 ± 309 µmol kg$^{-1}$ for the entire record (1997-2020, not shown). Compared to the Elbe Estuary, similar TA (< 2200 µmol kg-1) and DOC (< 450 µmol kg-1) concentrations were observed in an intertidal saltmarsh in the northeast USA, where the organic alkalinity fraction contributed a minimal amount of 0.9-4.3% to the TA (Song et al., 2020). Thus, the organic alkalinity could constitute only a small fraction of the TA in the Elbe Estuary. In future studies, either the difference between calculated and measured TA (Kuliński et al., 2014) or direct measurements of organic alkalinity (Song et al. 2020) could be used to quantify the organic alkalinity influence on TA in the Elbe Estuary.

## 4.1 The recent drought modulates estuarine carbon cycling

The recent drought period (2014–2020) has altered estuarine carbon cycling in several ways. Late spring (May) river discharge significantly decreased. By 2020, it reached levels (264 ± 19 m$^3$ s$^{-1}$) usually observed during summer and early autumn, thus extending the dry summer season into late spring. As a result, the estuarine water residence time increased compared to non-drought years by approximately 3 times (Table S16), using a function estimated by Bergemann et al. (1996), where a decrease from 700 m$^3$ s$^{-1}$ to 250 m$^3$ s$^{-1}$ would extend the residence time of the mid-estuary from 5 to 17 days. The significant decrease in discharge coincided with a significantly higher, up to double, internal DIC load in the mid to lower estuary (Table S9–S10). This is unexpected, and indicates that low discharge in the Elbe River in May allows for a longer remineralization period of POC and DIC production in the mid–lower Elbe Estuary. We deduce that this was most likely enhanced by the high and sufficient POC loading during the growing season, specifically to the mid-region, coupled with increased water residence time in the mid–lower estuary regions.

In contrast to late spring, the summer internal DIC load in the mid and lower estuary was significantly lower, by 35–42%, during the recent drought period between 2014 to 2020 (June–

July), compared to the non-drought period (Table S9–S10), where floods were excluded to allow comparisons between the non-extreme situation and the drought event. For the last 3 years of the recent drought (2018–2020), we observed a shift in the ecosystem parameters, with a decrease in pH down to 7.7 and an increase to positive AOU in the upper estuary (zone 1), indicating a shift to dominating heterotrophy (Fig. 2). Findings of Kamjunke et al. (2022) confirm the efficient decomposition of algal organic carbon in the upper Elbe Estuary (585 Elbe-km, z1) during the drought in September 2019. Schulz et al. (2023) reported nitrate depletion, down to 0.2 µM in this region (585 Elbe-km), indicating nutrient limitation of primary production in late July and August 2018–2019. Furthermore, the FGG Elbe observed the oxygen depleted zone extended to the upper estuary and further upstream in 2018 (Gregor Ollesch, pers. comm.). This opposes the long-term trend of the upper estuary (z1) with dominating autotrophy in late spring and summer, also shown in Amann et al. (2015). An extended drought period over several years, like the one observed in the Elbe Estuary, could result in an eventual shift in carbon processing, with POC decomposition and DIC production further upstream during summer, which can contribute to a decreasing internal DIC load in the mid-lower estuary. The DIC produced as $CO_2$ may not be buffered by the carbonate system in the upper estuary due to the absence of seawater containing $CO_3^{2-}$ influence. Overall this could lead to larger $CO_2$ release to the atmosphere, starting with the upper estuary and could lower the along-estuary DIC export to the coast. This suggests that prolonged droughts significantly impact carbon cycling and ecosystem functioning, modulating the function of an estuary as carbon source or sink to the atmosphere and the coast. This also indicates a non-linear response of the ecosystem to forcing due to climate change. The potential changes in the magnitude of $CO_2$ evasion in the upper estuary during the summer drought period should be further investigated.

**4.2 Controls on inorganic carbon in the lower-outer estuary**

We find that – in contrast to the situation in the mid-estuary – POC loading from the upper estuary cannot account for the internal DIC load in the lower estuary, due to a significant surplus of internal DIC load compared to POC load in the upper estuary (Table S13), by an average of 1.3–1.9 times (Table S12). In the outer estuary, negative AOU and higher pH indicated net autotrophy in late spring and summer. This coupled with elevated POC, reaching 16% of SPM (Fig. S9), highlights increased availability of labile OM. We observed mainly positive (42%), and few negative (13%), deviations of DOC from conservative mixing in May to August

between 1997 and 2020 (Figs. S10–S13), also suggesting an addition of OC in the lower and outer region. Higher TA compared to DIC in the outer estuary, coupled with lower $pCO_2$ levels, closer to the respective annual mean atmospheric partial pressure (Fig. 4), corroborate the idea of net autotrophy in the outer estuary, e.g. as described in Brasse et al. (2002). The significant positive correlation between DIC and AOU, and negative correlation with pH, in summer in the outer estuary indicate a prevalent DIC drawdown in the outer estuary, with such correlations also found by Reimer et al. (1999) in the German Bight. However, rapid *in situ* remineralization of autochtonous organic carbon from the coastal region, rather than POC from the upper estuary, could counteract a strong DIC depletion as observed in the upper estuary. For example, Reimer et al. (1999) reported 60–75% of the autochtonous primary produced biomass was remineralized in the surface layer of the German Bight. This newly produced OM may also be transported into the lower estuary during flood tide, and undergo remineralization subsequently producing DIC, as proposed by Voynova et al. (2015) for the area between the Delaware Bay and Murderkill Estuary. This potential source of DIC from marine OC in the lower to outer estuary was also indicated by predominate positive excursions in DIC from conservative mixing lines in May to August between 1997 and 2020 (Figs. S4–S7), and suggested by Schulz et al. (2023).

Besides internal production of DIC, Hoppema (1993) and Voynova et al. (2019) conclude that remineralization of OM within Wadden Sea sediments and subsequent DIC and TA release into the water column considerably contribute to elevated DIC concentrations in adjacent coastal regions. The Wadden Sea coastal region, adjacent to the outer Elbe Estuary, receives around 100 g C m$^{-2}$ yr$^{-1}$ OM from the North Sea (van Beusekom et al., 1999). Voynova et al. (2019) also found largest TA generation in summer and autumn at 7.8-8°E, west of the outer estuary (8.3-8.5°E, Fig. 1a), reaching 2400 µmol kg$^{-1}$ in summer 2017, exceeding summer DIC (2250 ± 50 µmol kg$^{-1}$) while similar to summer TA (2491 ± 71 µmol kg$^{-1}$) in the outer estuary. Therefore, the remineralization of OM in Wadden Sea tidal flats exported to the coastal region and facilitated by tidal flow could contribute to the enhanced DIC in the outer estuary, especially during the peak DIC in summer.

Freshwater and saltwater marshes are also adjacent to the inner and outer Elbe Estuary and are known to contribute to the estuary DIC budget (Weiss, 2013). Weiss (2013) estimated the DIC export from the tidal marshes can account for 2.8–10.2% of the mean annual DIC from the Elbe Estuary at 63.5 ± 1.4 Gmol yr$^{-1}$ (Amann et al., 2015). The total internal DIC load of 6.4 Gmol

C month$^{-1}$ for May to August between 1997 and 2020 (Table S12) represents 7–20% of the annual DIC export to coastal waters (Fig. 6). This highlights the importance of accounting for different carbon sources to disentangle the mechanisms responsible for carbon turnover in this region and to help improve regional and global carbon budget calculations.

**4.3 Tentative inorganic carbon export estimates**

The inorganic carbon in estuaries eventually settles in sediments, is released to the atmosphere as $CO_2$ or exported to the adjacent coastal waters (Kempe, 1982). Amann et al. (2015) deduced

that the latter two were the major export pathways in the Elbe Estuary. We have estimated the annual DIC water-air fluxes and lateral flux during the recovery state (Rewrie et al. 2023), and have used the sum of the two as the estimated total export of DIC out of the Elbe Estuary.

Out of the total Elbe Estuary DIC export, between 77 and 94 %, up to 89 ± 4.8 Gmol C yr$^{-1}$,

was laterally transported to coastal waters, and between 6 and 23 % was released via $CO_2$ evasion to the atmosphere, and thus only up to 10 Gmol C yr$^{-1}$. This matches the ratio between DIC export to the atmosphere and to the coastal area quantified by Amann et al. (2015). Also, the water-air $CO_2$ flux range (Fig. 6) places the Elbe Estuary within the range of flux estimates for North Sea tidal estuaries (Volta et al. 2016). Amann et al. (2015) suggested the water

residence time, not only influenced the $CO_2$ flux to the atmosphere, but also the magnitude of DIC exported to the adjacent coastal waters. Based on the long-term mean annual Elbe River discharge (Fig. 2a), the Elbe Estuary was characterized by an average residence time of around 3 weeks estimated by Bergemann et al. (1996) as shown in Table S16. The Satilla River estuary in the US was characterized by a longer average residence time of 8 weeks, and Cai and Wang

(1998) calculated that 90 % of the total DIC export was $CO_2$ evasion to the atmosphere. In contrast, only of 4.6 % of the DIC export from the Changjiang River estuary in East China was released to the atmosphere, which features a shorter residence times of a week or less (Zhai et al. 2007). This confirms that the carbon cycling in estuarine and coastal waters is highly dependent on hydrological conditions.

Despite the significantly higher internal DIC load in late spring during the drought period (2014–2020), the lowest DIC export to coastal waters of 38 ± 5.4 Gmol C yr$^{-1}$ occurred during the drought, and this was a 24% decrease compared to the non-drought period, when excluding flood events. The significantly lower internal DIC load in summer and the 32% decrease in the

mean annual Elbe River discharge likely modulated the decrease in the annual DIC export. Severe drought conditions have previously resulted in smaller carbon exports from estuaries to the ocean (Tian et al., 2015; Cavalcante et al., 2021). For instance, in the Mississippi River basin, all C fluxes (DIC, DOC and POC) decreased by 38% to lowest fluxes in the 2006 dry year relative to a 10 year average (Tian et al., 2015). Major changes in the river discharge, such as significant reduction during prolonged drought, are therefore likely to have an impact on inorganic carbon delivery to the coastal ocean, and therefore climate change related disruptions seem to have major impact on the estuary-coast carbon budget.

**5 Conclusion**

To assess the impact on the estuarine ecosystem, we compare the processes over the period of 1997 to 2020, described as the recovery state by Rewrie et al. (2023). It followed major shifts in ecosystem state after the 1980s heavy pollution. The significant increase in DIC in the mid Elbe Estuary of 6–21 µmol kg$^{-1}$ yr$^{-1}$ from 1997 until 2020, during late spring and summer, is associated with a concomitant significant increase in POC, at 8–14 µmol kg$^{-1}$ yr$^{-1}$, in the upper estuary. The observed POC increase in the upstream waters was related to an overall improvement in water quality, with a significant decrease in BOD$_7$ by over half since 1997. The significant positive correlation between the along-estuary DIC gain in the mid-estuary and POC in the upper estuary (1997–2020) indicates that the amount of POC from the upper estuary is sufficient to drive the long-term DIC increase in the mid-estuary. The decomposition of POC by the mid-estuary region, prior to export to adjacent coastal waters also shows that the estuary is an efficient filter for upper estuary POC inputs.

A notable environmental driver in modulating carbon cycling in the estuary is a multi-year drought between 2014 and 2020, with significantly lower mean Elbe River discharge at 468 ± 234 m$^3$ s$^{-1}$. We find that the drought extended the dry season into late spring, lengthening the water residence time by approximately 3 times. The increased residence time allowed for a longer remineralization period for POC in May, resulting in more than doubling the internal DIC load in the mid-lower estuary, compared to the non-drought period (1997–2013). Coupled with the high POC loading from the upper estuary, this resulted in the highest internal DIC load in the mid estuary region.

In the lower to outer estuary, we find that different mechanisms likely support the maximum DIC concentrations observed here. The internal DIC load in the lower estuary was on average 1.3–1.9 times higher than the POC load from the upper estuary. We therefore postulate that allochtonous OM from adjacent coastal regions or autochthonous (i.e. phytoplankton) labile outer estuary OM was transported into the lower estuary and remineralized there, supporting higher DIC concentrations. Additionally, the export of OM from the surrounding Wadden Sea sediments followed by remineralization within the coastal waters (Voynova et al., 2019) and import of DIC from adjacent tidal marshes (Weiss, 2013) are mechanisms that could explain high outer estuary DIC concentrations. To accurately quantify DIC production, and determine the magnitude and direction of DIC loads in the lower and outer estuary, a study should concentrate on nearshore waters, whereby both flood and ebb tide must be considered.

On an annual basis, the Elbe Estuary acts as a source for $CO_2$ to the atmosphere, with an estimated maximum of 10 Gmol C $yr^{-1}$ released to the atmosphere and a maximum of $89 \pm 4.8$ Gmol C $yr^{-1}$ exported to adjacent coastal waters. The ratio between DIC export to the atmosphere and to the coastal area varied between 1997 and 2020, with 77–94% laterally transported to coastal waters, and only a maximum of 23% released to the atmosphere, making the estuary an efficient system to fix carbon as DIC in its outflow to the coastal regions. During a 6 year drought, the DIC export to coastal waters decreased significantly by 24% relative to the non-drought period, down to $38 \pm 5.4$ Gmol C $yr^{-1}$, and this shows the major effects of climate change on river discharge, and on the timing and magnitude of inorganic carbon export. The drought had no apparent influence over the estimates of water-air $CO_2$ flux. The DIC production in and export from the Elbe Estuary is, therefore, an important factor in North Sea and land-to-sea carbon budgets. While we have only provided estimates of the changes in carbon export, we show that it is essential to take into account seasonal, but also long-term changes in the DIC production and consumption within an estuary. This knowledge is crucial for predicting carbon cycling at the land to ocean continuum, such as long-term changes in water-air $CO_2$ flux, DIC export to coastal waters, as well as the impacts of prolonged droughts on the coastal carbonate system.

*Data availability.* The data for DIC, POC and the ecosystem parameters are freely available from the data portal of the Flussgebietsgemeinschaft Elbe (FGG, River Basin Community; https://www.fgg-elbe.de/elbe-datenportal.html). The daily mean ambient air pressure and wind speed data are available from E-OBS meteorological data for Europe from the Copernicus Climate Data Store; https://cds.climate.copernicus.eu. The atmospheric $CO_2$ data are available from the Global Monitoring Laboratory; https://gml. noaa.gov/ccgg/trends/gl_data.html.

*Author contributions.* LR and YV designed the concept of the study, with contributions from BB. LR conducted the data analysis and evaluation, with support by YV. LR led the writing with contributions from YV. BB, JB, AK and GO contributed to scientific input and revisions of the manuscript. All authors approved the final submitted manuscript.

*Competing interests.* The authors declare that they have no conflict of interest

*Acknowledgements.* This research was funded by the CARBOSTORE" project funded by the German Federal Ministry of Education and Research, BMBF, the EU project DANUBIUS-IP (Grant agreement 101079778) and the Helmholtz Association funding program 'Changing Earth'. We thank Ulrich Wiegel for providing information on the carbon measurement methods at FGG Elbe. We are grateful to the researchers and staff at FGG Elbe for the collection of DIC and ecosystem samples. We thank Kirstin Dähnke and Gesa Schulz for providing detailed information on the nitrogen cycling in the upper estuary and helpful discussions. Thanks to Vlad Macovei for providing information on using Ocean Data View. We thank the associate editor Tyler Cyronak and the editorial support team at Copernicus Publications. We thank the two anonymous reviewers whose helpful comments improved the manuscript.

*Financial support.* This work was supported by the CARBOSTORE" project funded by the German Federal Ministry of Education and Research, BMBF, the EU project DANUBIUS-IP (Grant agreement 101079778) and the Helmholtz Association funding program 'Changing Earth'.

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
