# Peer review of "Recent inorganic carbon increase in a temperate estuary driven by water quality improvement and enhanced by droughts"

_EGUsphere, 2023_

## Author Comment (AC1)

**Response to reviewers:**

**We thank both reviewers for their constructive and helpful review of our paper.**

**From both reviewers there were queries concerning the reference 'Rewrie et al. (in review)'. This manuscript was accepted by Limnology and Oceanography with the Article DOI: http://doi.org/10.1002/lno.12395 and is now published. We are happy to provide a brief explanation on the differences between the two papers.**

**Rewrie et al. (2023) assessed changes in DIC and ecosystem parameters from 1985 to 2018. From the abstract: 'Based on an extensive evaluation of key ecosystem variables, and an analysis of the available inorganic and organic carbon records, this study has identified three ecosystem states in recent history: the polluted (1985-1990), transitional (1991-1996) and recovery (1997-2018) states. The polluted state was characterised by very high dissolved inorganic carbon (DIC) and ammonium concentrations, toxic heavy metal levels, dissolved oxygen (DO) undersaturation and low pH. During the transitional state, heavy metal pollution decreased by > 50%, and primary production re-established in spring to summer, with weak seasonality in DIC. Since 1997, during the recovery state, DIC seasonality was driven by primary production, and DIC significantly increased by 11 μmol L$^{-1}$ yr$^{-1}$, and > 23 μmol L$^{-1}$ yr$^{-1}$ in the recent decade (2008-2018), in the mid to lower estuary, indicating that, along with the improvement in water quality the ecosystem state is still changing'**

**In the present manuscript, we focus on the changes in DIC in the recent recovery state (1997-2018) and extend the dataset by two years to 2020 due to more recent data availability. This study investigates the reason for the DIC increase over time by utilizing changes in organic carbon in the upper estuary. This study also evaluates the impact of the recent drought on the carbon cycling in the Elbe Estuary. We believe that publishing the manuscript described above, which precedes this one will answer most questions raised by the reviewers. However, we have attempted to carefully address the reviewer's comments in the sections below.**

Comments to Authors

This is an interesting dataset describing longitudinal gradients and long-term trends in POC and DIC for the Elbe Estuary. The findings from an analysis of these data would be of interest to estuarine ecologists and to the broader community studying the global C cycle. My main concern with the paper is that there are many aspects that are either not well explained, or insufficiently explained. This issue pervades key components of the paper including conceptualization of the study, description of methods and inferences made from the data.

Methods: the level of detail in explaining methods is highly uneven among the various components of the study. For example, the means for calculating air-water CO2 fluxes is highly detailed, whereas other equally important elements (e.g., POC loads and DIC mineralization) are hardly described at all in the main body of the paper. These fluxes, and how they are derived should be explained more fully and at an early stage. For example, in the Abstract there is reference to the "spring internal DIC load" (line 30), but at this stage, the leader is not likely to understand what this is (remineralization of organic C) or how it was determined. Also, I did not see an explanation of how POC was measured (perhaps I missed this).

**To ensure the methods are clearly provided we will move equations S2 to S5 from the supplementary materials into the methods section.**

**We had referenced Rewrie et al. (2023) which extensively describes the inorganic and organic carbon measurement methods to reduce repetition. However, to increase clarity in this manuscript, we will include a sentence on line 134 to describe that POC was calculated as the difference between TOC and DOC (Table S1), with an estimated uncertainty of 20% based on the Pythagorean Theorem (U. Wiegel, pers. comm).**

Introduction: the first paragraph focuses on C cycling in estuaries, with a mention of eutrophication. The second paragraph focuses of climate change, and specifically, the occurrence of drought. The third (final) paragraph does not link the ideas presented in the first and second paragraphs in such a way as to provide a clear direction for the paper. There should be some consideration (prediction) of how estuarine C cycles may be affected by drought. My initial reaction was that drought would reduce external (watershed) inputs of POC and DOC to the estuary. I was surprised that there was no consideration here, or elsewhere, of the importance of allochthonous organic matter inputs, particularly as this is the driving mechanism accounting for the excess of C mineralization relative to autotrophic fixation in estuaries (see e.g., Hoellein et al. 2013). The third paragraph should provide some clear expectations of the direction of the paper from a conceptual point of view, which the stated objectives fail to do. Also, there should be some consideration of allochthonous C inputs and how these may have changed over time (and in response to drought).

**Thank you for this suggestion and reference. We do in fact argue in the later sections of the manuscript that allochtonous POC sources are key to the DIC processing in the estuary. To highlight the importance of allochthonous C in the introduction, we will edit the first paragraph on line 50-53. E.g. 'River-borne and in situ primary production supplies allochthonous and autochthonous organic carbon to and within estuaries (Abril et al. 2002; Hoellein et al. 2013), subsequently providing labile forms of carbon'.**

**In the second paragraph, we described the influences of extreme floods and droughts on OM and nutrient cycling within estuaries on lines 71-76. To link the first and second paragraph we will change the sentence on lines 74-76: 'This in turn can extend the retention of carbon and nutrients during droughts, permitting more extensive remineralization of allochthonous and autochthonous organic material within an estuary (Hitchcock and Mitrovic, 2015), and subsequently altering carbon and nutrient cycling'.**

**To clarify the reasons behind assessing the period between 1997 and 2020 we will change the third paragraph on lines 87-88 to 'To assess the impact of the drought on the carbon cycling in the estuarine ecosystem, we assess a longer period between 1997 and 2020, to allow comparisons between a non-drought and drought period.'**

The reliance on Rewrie et al. in review, here, and at many points throughout the paper, is not helpful, particularly as these are vague references to "ecosystem recovery", "major shifts in ecosystem state" and "amelioration of water quality". It is not clear to the readers of this paper what these changes are, and what implications they may have for C cycling in the Elbe. There is a subsequent statement that these changes include a reduction in BOD and an increase in NPP. The former implies that the changes may

have to do with improved wastewater treatment resulting in reduced organic matter inputs to the estuary.  But if that is the case, should these fluxes not be accounted for in this paper describing trends in POC and DIC?  Also, if wastewater treatment practices have been improved, this should bring about a reduction in nutrient inputs, and potentially diminish, not enhance NPP.  I short, I found it difficult to understand the long-term trends presented in this paper while not understanding what are the changes occurring in this system that seem to be the focus of a different paper.

**To clarify the key findings of Rewrie et al. (2023) we will change the introduction from line 88 and will remove 'It has been described as the ecosystem recovery state of the estuary (Rewrie et al., 2023), following major shifts in the ecosystem state after the 1980s heavy pollution.' And then include 'Since 1997, the ecosystem of the Elbe Estuary was designated in a recovery state (Rewrie et al., 2023), characterised by non-toxic levels of heavy metals permitting autotrophy and heterotrophy within the estuary, which followed a heavily polluted state in the 1980s and the ensuing transitional state (1991-1996).'**

**To clarify the changes in the water quality in the Elbe Estuary we will specify the BOD changes observed in the upper estuary. Since the decrease was not directly inverse to the POC increase, we will remove 'opposite of the observed POC increase' and include on line 410 'The summer mean $BOD_7$ decreased from 12±1.7 mg $L^{-1}$ in 1997–2005 to 8±1.1 mg $L^{-1}$ in 2006–2020'. After this sentence we will also include 'While there was a continuous decrease in nutrients from the late 1990s (Wachholz et al. 2022), the nutrient supply was sufficient to support phytoplankton production (Kamjunke et al., 2021; Dähnke et al 2022).' This highlights that despite the overall decrease in nutrients the concentrations were predominately sufficient for primary production in the Elbe River and Estuary, as has been found by other researchers (e.g. Dähnke et al 2022), who will also be referenced.**

**Dähnke, K., Sanders, T., Voynova, Y., & Wankel, S. D. (2022). Nitrogen isotopes reveal a particulate-matter-driven biogeochemical reactor in a temperate estuary. Biogeosciences, 19(24), 5879-5891.**

**Wachholz, A., Jawitz, J. W., Büttner, O., Jomaa, S., Merz, R., Yang, S., & Borchardt, D. (2022). Drivers of multi-decadal nitrate regime shifts in a large European catchment. Environmental Research Letters, 17(6), 064039.**

**We would like to highlight that in the submitted manuscript we had indeed identified a potential reduction in nutrient inputs in 2018 to 2019 on lines 508-510, which was associated with the drought.**

Results: throughout the paper, loads are presented as mass per unit of time (e.g., the total mass of CO2 leaving the estuary), which is not very helpful to facilitating cross-system comparisons (vs. presenting these as values per unit area of the estuary).  By analogy, river loads (watershed export) are more commonly normalized to watershed area (i.e., as a yield per square meter) to allow comparisons among watersheds of different size.  Readers could take the values provided in this paper and divide by the specified area of the estuary to obtain estimates for comparisons to other estuaries. But the potential for making inter-system comparisons would be enhanced if the authors were to present their data as per unit area of the estuary.

**Thank you for this suggestion. The reason why we used the mass per unit of time was to allow us to calculate a ratio of the export and compare to other estuaries on**

**lines 582-587. To allow the reader to see our results in both forms of units we will combine Gmol yr$^{-1}$ to one axis with a log scale and will also present on the twin axis the export plot per unit area.**

Other Comments:

The Abstract lacks a 'big picture' perspective. The overall findings of the study are difficult to discern among the details of the results.

**To put our results in perspective to other studies we will include (italic) and change to the following:**

**'We have identified that seasonal changes in DIC processing in an estuary require consideration in order to understand *and accurately estimate* the long-term and future changes in air-water CO$_2$ flux *and* DIC export to coastal waters. , as well as the impacts of prolonged droughts on the land-ocean carbonate system.'** *Regional and global carbon budgets should take into account carbon cycling in estuaries, in relation to impacts of water quality changes and extreme hydrological events'*

**The aim here was to describe that the changes in DIC flux from estuaries will change due to what DIC is added in the estuary, which itself is influenced by changes in availability of labile POC (water quality) and the influence of external forces, in this study hydrological event (drought) influences both inside DIC addition and external flux.**

Site Description: it would be helpful if this included an indication of salinity levels along the length of the estuary (perhaps add data to Figure 1, or at least delineate polyhaline, mesohaline, etc.).

**Thank you for this suggestion. To address this we will include an average with standard deviation salinity gradient on Figure 1b.**

DOC: I was surprised that in a paper on estuarine C dynamics there was virtually no mention of DOC. Is it the case that internal production of POC and subsequent remineralization of POC are the dominant C fluxes in this system? At a minimum, it would be useful to report the proportions of total C represented by DIC, POC and DOC in river inputs to the estuary vs. relative contributions in export to the sea (and for drought vs. non-drought conditions).

**Yes, we suggest that the production of allochtonous POC (in the river and in the coastal regions adjacent to the estuary) and its subsequent remineralization of POC are controlling the C fluxes in this system, and are responsible for the recent observed increase in DIC. We will include a reference in the introduction describing the respiration of organic carbon in the Elbe Estuary, for example, 'The organic carbon cycling in the Elbe Estuary was evaluated before (Amann et al., 2012), identifying that from the late 1990s, POC fuelled heterotrophic respiration whereas respiration of DOC in the estuary was negligible. However, the last decade was not included.'**

**In response to both reviewers, we will include information on DOC to show the concurrent DOC changes as well, to support our findings that the remineralization of POC, rather than DOC fuels the DIC production. To address the reviewer's**

concern we will estimate the removal of DOC and POC in the estuary, defined as filtering capacity in Amann et al. (2012) in the Elbe Estuary.

To assess the importance of the upper estuary DOC and POC respiration and subsequent DIC production in the Hamburg Harbour and the mid-estuary the organic carbon removal as shown in Amann et al. (2012) expressed in percent was calculated:

$$C_{FC}\% = \frac{(C_{z1} - C_{zi})}{C_{z1} \times 100}$$

Where C is the POC or DOC concentration in the respective zone with zi representing zones 2-3 for POC and zones 2-5 for DOC. The OC removal for POC was calculated for zone 2 and 3 due to the influence of the maximum turbidity zone in zone 4 and 5 (Amann et al. 2012). The negative values indicate OC addition.

We found that the POC removal was up to ~ 4 times greater (80%) compared to the DOC removal (21%). In the regions when DOC and POC were removed between 1997 and 2020, the mean removal was 7 ± 5% and 41 ± 18%, corresponding to a mean concentration loss of 39 ± 30 µmol kg$^{-1}$ and 160 ± 104 µmol kg$^{-1}$, respectively. This indicates respiration of upper estuary POC dominates DIC production in the Hamburg Harbour to mid Elbe Estuary.

To further support upper estuary POC dominates DIC production in the mid-estuary, we would like to highlight that in the submitted article we discussed (on lines 429-430) 'The magnitude of along-estuary DIC gain in the mid-estuary and POC input into the estuary show no significant difference in late spring and summer (Table S5).' We would also like to highlight in the submitted article we discussed POC in % of SPM was used to describe the mineralisation of POC in the mid-estuary. On line 434 'We find that POC drops to < 4% of SPM in May to August (1997–2020) in the mid estuary (z4–z5, Fig. S5), indicating widespread OM remineralization in the estuary.'

The story that the remineralisation of upper estuary POC dominates DIC production in the Hamburg Harbour to the mid-estuary does not change. We will include figures for the OC removal in percent and the concentrations of DOC and POC along the estuary for late spring (May) and summer (June-August) in the results and supplementary material and integrate the findings into the discussion.

In the lower to outer estuary, we have identified that other sources of OM likely support DIC production as described in the submitted article on lines 525 to 542. To assess DOC production in this region, we assessed the mixing of DOC along salinity for May to August between 1997 and 2020. We found positive non-conservative mixing of DOC along the salinity gradient in 42% of the assessed months. This corroborates our suggestion of OC production in the outer estuary can fuel DIC production therein. We will include additional DOC mixing along the salinity gradient figures in the supplementary material to support our findings in the discussion.

Amann, T., Weiss, A., & Hartmann, J. (2012). Carbon dynamics in the freshwater part of the Elbe estuary, Germany: Implications of improving water quality. Estuarine, Coastal and Shelf Science, 107, 112-121.

Results (line 240): there is frequent use of indirect metrics (AOU, pH) to make inferences about autotrophic activity.  Are there no primary data that can be used to support these inferences (e.g., CHLa measurements)?

**To support primary production in spring and summer we plotted a seasonal plot of the monthly mean chl-a at 585.5 Elbe-km calculated from 1997 to 2020. We will place this plot in the supplementary material, since it is only supporting the findings of AOU and pH in the upper estuary, without altering the manuscript findings. AOU we find is a more powerful measurement of production/respiration balance. A time-series of chl-a does not provide more evidence than POC, which was chosen to represent the labile carbon produced during primary production and available for remineralisation, and for which we found a strong and significant correlation with DIC, in the same order of magnitude, and already in units of carbon. We will include (in italics) on line 242 following 'This suggests that dominating autotrophy depletes DIC in the upper estuary, and most likely the upstream river regions, *which is supported by highest chlorophyll a concentrations in May to August at 585.5 Elbe-km, reaching 166 ± 74 µg L$^{-1}$.'***

Results (line 260): the statement "significant POC increases occurred..." is followed by some specified values, but it is unclear what these numbers represent (the mean concentration? the increase in concentration?  If the latter, increase relative to what?).

**To clarify this sentence we would change to 'Significant mean POC increases occurred in late spring (May, 14 µmol C kg-1 yr-1) and summer (June–August, 8 µmol C kg-1 yr-1) in the upper estuary (Fig. 3, Table 1) from 1997 to 2020'.**

Results (line 278): I did not understand why the TA:DIC ratio should be of interest, or what is the significance of this ratio being <1.

**To clarify the importance of using the TA: DIC ratio we will include the following reference on line 276: The ratio of TA to DIC can serve as a broad indicator of the sources of carbon, for example, when < 1 this can reflect DIC input in the form of CO$_2$ (Joesoef et al., 2017).**

**Joesoef, A., Kirchman, D. L., Sommerfield, C. K., & Cai, W. J. (2017). Seasonal variability of the inorganic carbon system in a large coastal plain estuary. Biogeosciences, 14(21), 4949-4963.**

Discussion (line 425): do you mean to say that mineralization rates increase linearly with POC concentrations, or that mineralization efficiency increases (i.e., that the proportion of POC that is remineralized increases)?

**Upon feedback from both reviewers, to clarify we will change this section from line 423 to 427 to 'The upper estuary POC in late spring and summer tripled since the onset of the recovery state in 1997, which we suggest is driven largely by allochtonous POC produced in the Elbe River and autochthonous POC produced in the upper estuary. Abril et al. (2002) reported that POC mineralisation efficiency (i.e., the percentage of POC mineralized) is a linear function of POC concentration, and considering the increased POC concentration, we can expect a higher turnover of POC in the estuary in recent years. That is, from 1997 to 2020, the increase in POC in the upper estuary enhanced the availability of POC for remineralisation, and subsequently increased DIC production, as was observed in the increase in DIC concentration in the Elbe Estuary (Fig 3).'**

Table 3.2: it would be helpful to include the standard error of the slope.

**We will include the standard error of the slope for DIC and TA in Table 1.**

Figure 4: it is somewhat confusing to use the designation "m-1" as this is much more commonly used to indicate per meter. Perhaps "mon-1" would be the better abbreviation for monthly values?

**Thank you for highlighting this. We will change the unit to mon$^{-1}$ in the figure and month$^{-1}$ in text.**

---

## Author Comment (AC2)

**Response to reviewers:**

**We thank both reviewers for their constructive and helpful review of our paper.**

**From both reviewers there were queries concerning the reference 'Rewrie et al. (in review)'. This manuscript was accepted by Limnology and Oceanography with the Article DOI: http://doi.org/10.1002/lno.12395 and is now published. We are happy to provide a brief explanation on the differences between the two papers.**

**Rewrie et al. (2023) assessed changes in DIC and ecosystem parameters from 1985 to 2018. From the abstract: 'Based on an extensive evaluation of key ecosystem variables, and an analysis of the available inorganic and organic carbon records, this study has identified three ecosystem states in recent history: the polluted (1985-1990), transitional (1991-1996) and recovery (1997-2018) states. The polluted state was characterised by very high dissolved inorganic carbon (DIC) and ammonium concentrations, toxic heavy metal levels, dissolved oxygen (DO) undersaturation and low pH. During the transitional state, heavy metal pollution decreased by > 50%, and primary production re-established in spring to summer, with weak seasonality in DIC. Since 1997, during the recovery state, DIC seasonality was driven by primary production, and DIC significantly increased by 11 $\mu$mol L$^{-1}$ yr$^{-1}$, and > 23 $\mu$mol L$^{-1}$ yr$^{-1}$ in the recent decade (2008-2018), in the mid to lower estuary, indicating that, along with the improvement in water quality the ecosystem state is still changing'**

**In the present manuscript, we focus on the changes in DIC in the recent recovery state (1997-2018) and extend the dataset by two years to 2020 due to more recent data availability. This study investigates the reason for the DIC increase over time by utilizing changes in organic carbon in the upper estuary. This study also evaluates the impact of the recent drought on the carbon cycling in the Elbe Estuary. We believe that publishing the manuscript described above, which precedes this one will answer most questions raised by the reviewers. However, we have attempted to carefully address the reviewer's comments in the sections below.**

Rewrie et al. (2023) investigated the variations in biogeochemical parameters, including DIC, POC, DO, and pH, in the Elbe Estuary over a long time period of 23 years. They discovered a significant long-term increase in DIC in the Elbe Estuary, which might be primarily influenced by a rise in POC content in the upper estuary. The researchers suggest that the increased internal load of DIC during the drought period can be attributed to an extended residence time in the estuary, allowing for a longer period of POC remineralization. The manuscript highlights the correlation between POC in the upper estuary and DIC in the mid and lower estuary, which may provide valuable insights for understanding DIC variations driven by climate change and human-induced disturbances. However, I have several concerns regarding the manuscript, and I hope the authors will carefully consider them.

Main comments

1.The authors have solely examined the correlation between POC data in the upper estuary and DIC concentrations in the mid and lower estuary. Including POC data from the mid and lower estuary in the manuscript would offer a clearer understanding of the relationship between POC and DIC. Additionally, while the authors have provided a preliminary estimate of inorganic carbon export, they have not thoroughly discussed

other potential factors influencing inorganic carbon export. I am curious about the relationship between dissolved organic carbon (DOC) and DIC in the Elbe Estuary. Since labile DOC can also be consumed by microbes, it may contribute to the DIC pool. I recommend the authors add some discussions on the correlations between DOC and DIC.

**We thank the reviewer for this suggestion to include DOC.**

**We will include a reference in the introduction describing the respiration of organic carbon in the Elbe Estuary. For example after 'During the current recovery state, the annual mean DIC in the mid to lower Elbe Estuary has increased significantly by up to 11 µmol L$^{-1}$ yr$^{-1}$ from 1997 to 2018 (Rewrie et al., in press), but the source of this increase remains unclear.' We will include 'The organic carbon cycling in the Elbe Estuary was evaluated before (Amann et al., 2012), identifying that from the late 1990s, POC fuelled heterotrophic respiration whereas respiration of DOC in the estuary was negligible. However, the last decade was not included.'**

**To address the reviewer's concern we will estimate the removal of DOC and POC in the estuary, defined as filtering capacity in Amann et al. (2012) in the Elbe Estuary.**

**To assess the importance of the upper estuary DOC and POC respiration and subsequent DIC production in the Hamburg Harbour and the mid-estuary the organic carbon removal as shown in Amann et al. (2012) expressed in percent was calculated:**

$$C_{FC}\% = \frac{(C_{z1} - C_{zi})}{C_{z1} \times 100}$$

**Where C is the POC or DOC concentration in the respective zone with zi representing zones 2-3 for POC and zones 2-5 for DOC. The OC removal for POC was calculated for zone 2 and 3 due to the influence of the maximum turbidity zone in zone 4 and 5 (Amann et al. 2012). The negative values indicate OC addition.**

**We found that the POC removal was up to ~ 4 times greater (80%) compared to the DOC removal (21%). In the regions when DOC and POC were removed between 1997 and 2020, the mean removal was 7 ± 5% and 41 ± 18%, corresponding to a mean concentration loss of 39 ± 30 µmol kg$^{-1}$ and 160 ± 104 µmol kg$^{-1}$, respectively. This indicates respiration of upper estuary POC dominates DIC production in the Hamburg Harbour to mid Elbe Estuary.**

**To further support upper estuary POC dominates DIC production in the mid-estuary, we would like to highlight that in the submitted article we discussed (on lines 429-430) 'The magnitude of along-estuary DIC gain in the mid-estuary and POC input into the estuary show no significant difference in late spring and summer (Table S5).' We would also like to highlight in the submitted article we discussed POC in % of SPM was used to describe the mineralisation of POC in the mid-estuary. On line 434 'We find that POC drops to < 4% of SPM in May to August (1997–2020) in the mid estuary (z4–z5, Fig. S5), indicating widespread OM remineralization in the estuary.'**

The story that the remineralisation of upper estuary POC dominates DIC production in the Hamburg Harbour to the mid-estuary does not change. We will include figures for the OC removal in percent and the concentrations of DOC and POC along the estuary for late spring (May) and summer (June-August) in the results and supplementary material and integrate the findings into the discussion.

In the lower to outer estuary, we have identified that other sources of OM likely support DIC production as described in the submitted article on lines 525 to 542. To assess DOC production in this region, we assessed the mixing of DOC along salinity for May to August between 1997 and 2020. We found positive non-conservative mixing of DOC along the salinity gradient in 42% of the assessed months. This corroborates our suggestion of OC production in the outer estuary can fuel DIC production therein. We will include additional DOC mixing along the salinity gradient figures in the supplementary material to support our findings in the discussion.

Amann, T., Weiss, A., & Hartmann, J. (2012). Carbon dynamics in the freshwater part of the Elbe estuary, Germany: Implications of improving water quality. Estuarine, Coastal and Shelf Science, 107, 112-121.

2. The authors utilized the CO2SYS program to calculate TA and pCO2 using DIC and pH as input parameters. However, it should be noted that the TA model in the CO2SYS program does not account for the contribution of organic alkalinity. Previous studies have reported significant concentrations of organic alkalinity (ranging from 10 to 70 umol kg[-1]) in rivers and estuaries. Ignoring the presence of organic alkalinity in the calculated TA values may introduce substantial uncertainties. Therefore, I recommend that the authors include discussions addressing the uncertainties associated with the calculated TA values.

Thank you for this recommendation. We will include a discussion to compare the calculated TA and the available DOC measurements (from FGG Elbe data portal) with previous studies that describe organic alkalinity. The discussion example is; 'The CO2SYS program does not account for contribution of organic alkalinity in the calculated TA. Organic alkalinity can constitute a smaller fraction to TA compared to that provided by the inorganic compounds however, it could be a significant component of TA in systems influenced by dissolved organic matter inputs (Voynova et al., 2019). Mean DOC and TA in the Elbe Estuary (z1-z7) were respectively 498 ± 92 µmol kg[-1] and 1985 ± 309 µmol kg[-1] for the entire record (1997-2020, not shown). Similar TA concentrations (< 2200 µmol kg[-1]), but slightly lower DOC (< 450 µmol kg[-1]), were observed in an intertidal saltmarsh in the northeast USA. Song et al. (2020) reported the organic TA fraction contributed 0.9-4.3% of the TA, and thus only a minimal amount. In contrast, Hunt et al. (2011) reported a 21-100% contribution of organic alkalinity to TA, with much lower TA (116 µM to 956 µM) and extremely high DOC, up to 1480 µmol L[-1], in 15 rivers of northern New England (USA) and New Brunswick (Canada). Kuliński et al. (2014) reported the much lower TA in the river could explain the larger organic alkalinity contribution. Thus, it is likely organic alkalinity constituted a small fraction of the TA in the Elbe Estuary. There were changes in DOC along the upper to mid-estuary, with OC changes in percentage ranging between -124% and 21% in zones 2 to 5 between 1997 and 2020 (reference to OC removal figures and DOC concentration along the estuary). This indicates the potential contribution of DOC to organic alkalinity along the upper to mid Elbe Estuary. In the lower to outer estuary, there were positive non-conservative mixing of DOC along the salinity gradient and this may

have influenced production of organic alkalinity. However, in order to determine the amount of organic TA contributing to overall TA, the difference between calculated and measured TA or direct measurements organic alkalinity are required. In future studies, where both measured and calculated TA are available, we suggest the organic alkalinity influence on TA in the Elbe Estuary should be assessed.'

Hunt, C. W., Salisbury, J. E., & Vandemark, D. (2011). Contribution of non-carbonate anions to total alkalinity and overestimation of pCO 2 in New England and New Brunswick rivers. Biogeosciences, 8(10), 3069-3076.

Kuliński, K., Schneider, B., Hammer, K., Machulik, U., & Schulz-Bull, D. (2014). The influence of dissolved organic matter on the acid–base system of the Baltic Sea. Journal of Marine Systems, 132, 106-115.

Song, S., Wang, Z. A., Gonneea, M. E., Kroeger, K. D., Chu, S. N., Li, D., & Liang, H. (2020). An important biogeochemical link between organic and inorganic carbon cycling: Effects of organic alkalinity on carbonate chemistry in coastal waters influenced by intertidal salt marshes. Geochimica et Cosmochimica Acta, 275, 123-139.

Voynova, Y. G., Petersen, W., Gehrung, M., Aßmann, S., & King, A. L. (2019). Intertidal regions changing coastal alkalinity: The Wadden Sea-North Sea tidally coupled bioreactor. Limnology and Oceanography, 64(3), 1135-1149.

3.This manuscript extensively references a paper (Rewrie et al., in review) that is currently under review. It is generally not considered appropriate to rely heavily on the findings of a paper that is still undergoing the review process. Additionally, it seems there is a high correlation between that paper and the current manuscript. It would be helpful if the authors could provide an introduction outlining the differences and commonalities between this manuscript and the paper they have cited (Rewrie et al., in review) in the response to reviewers.

We have explained the Rewrie et al. (2023) manuscript above.

To clarify the key findings of Rewrie et al. (2023) we will change the introduction from line 88 and will remove 'It has been described as the ecosystem recovery state of the estuary (Rewrie et al., in review), following major shifts in the ecosystem state after the 1980s heavy pollution.' And then include 'Since 1997, the ecosystem of the Elbe Estuary was designated in a recovery state (Rewrie et al., in press), characterised by non-toxic levels of heavy metals permitting autotrophy and heterotrophy within the estuary, which followed a heavily polluted state in the 1980s and the ensuing transitional state (1991-1996).'

We will change the second aim to be specific '(2) investigate how the onset of the recent drought has modulated the carbon cycling within the estuary.'

4.The authors have utilized potentiometric pH and DIC measurements to calculate pCO2. However, it is important to note that potentiometric pH measurements may introduce significant uncertainties when measuring pH in saline waters. What is the salinity of the samples collected in the outer estuary? The authors should take into consideration this potential source of uncertainty in their analysis and discussion.

**On line 160, we applied the recommended total standard uncertainty for pH of 0.01 units. This error was also suggested in Dickson (1993). We will include this reference as well.**

**Dickson, A. G. (1993). The measurement of sea water pH. Marine chemistry, 44(2-4), 131-142.**

Minor comments

Abstract:

L.25 Is it possible that a portion of the gained DIC can be attributed to the degradation of DOC?

**We believe DIC gained by respiration of DOC is negligible compared to POC remineralization as answered above.**

Introduction:

L.55 "source"?

**Thank you for highlighting this. We will change this to 'source'.**

L.45-60 Why do the authors use "OM" and "POC" interchangeably in this paragraph? Does "OM" include DOC?

**Our aim was not to use OM and POC interchangeably. We wanted to highlight that organic carbon was a form of organic material as highlighted on lines 52-53 'This phytoplankton generated organic matter (OM) input'**

**To clarify we are referring to organic carbon in the paragraph and upon feedback from the Referee_1, we will change the sentence on line 50-53. E.g. 'River-borne and in situ primary production supplies allochthonous and autochthonous organic carbon to and within estuaries (Abril et al. 2002; Hoellein et al. 2013), subsequently providing labile forms of carbon'.**

**We will change the sentence on line 57 to 'This reduces labile OM export to the adjacent coastal waters (Abril et al. 2002; Crump et al., 2017; Sanders et al., 2018)'.**

L.55 I suggest that the authors include an introduction discussing the contribution of DOC degradation to DIC.

**We will include a reference in the introduction describing the respiration of organic carbon in the Elbe Estuary as answered above.**

Method:

L.150 Please add a reference for the CO2SYS program.

**We will include the reference (Lewis and Wallace, 1998) after 'Using the CO2SYS program version 2.5 in Excel'.**

L.130 what is the precision of pH, DOC, and DIC measurements?

**We will provide the significant digits/indication of results in Table S1.**

**The correction factor was related to the monthly river discharge. To clarify, we will change the sentence to 'A correction factor to the monthly river discharge was applied to each estuarine zone (zones 1 to 6) to account for tributary inputs along the estuary (Amann et al., 2015).'**

L 220 In the study of an estuary, why did the authors use a Schmidt number (Sc) in freshwater? The gas transfer velocity parameter (k) described by Wanninkhof (2014) is generally applicable to the open ocean.

**The flux of $CO_2$ between water and atmosphere was calculated for zones 1 to 6. For the entire record (1997-2020) the mean salinity in zone 6 was 11 ± 5 psu and for zones 1 to 6 was 2 ± 4 psu. In Wanninkhof (2014) the coefficients for the Schmidt number for seawater is for 35 psu. Therefore, due to the relatively lower salinity range in the Elbe Estuary we applied the freshwater Schmidt number. We decided to use a Schmidt number (Sc) in freshwater to also allow for comparisons of the calculated flux of $CO_2$ between water and atmosphere with published studies on the Elbe Estuary (Norbisrath et al. (2022) and Amann et al. (2015)).**

**We did not compare different approaches to calculate k as the aim was to provide a tentative evaluation of the overall inorganic carbon export as stated on lines 380. To clarify this in the methods on line 201 we will include (italic) 'To tentatively estimate the inorganic carbon export dynamics, the flux of $CO_2$ between water and atmosphere in mol m$^{-2}$ d$^{-1}$ was estimated for each sampled station, in the upper to lower region ($z1$-$z6$, Fig. 1a), between 1997 and 2020 with the equation.'**

**Even though we have not used a correction factor for the k calculation (such as in Volta et al. 2016), the mean air-water $CO_2$ flux (21 ± 6 mol C m$^{-2}$ yr$^{-1}$, 1997-2020) places the Elbe Estuary within flux estimates for North Sea tidal estuaries and specifically the Elbe Estuary of 27.4 mol C m$^{-2}$ yr$^{-1}$ (Volta et al. 2016). As requested by the other reviewer we will include the air-water $CO_2$ flux in both mol C m$^{-2}$ yr$^{-1}$ and Gmol yr$^{-1}$.**

Result:

L.285 How does the physical mixing between freshwater and seawater influence DIC in the outer estuary?

**In May to August, we found positive and negative non-conservative mixing in DIC along the salinity gradient in 43% and 23% of the assessed months between 1997 and 2020. This was not included in the manuscript however, we acknowledge that the non-conservative mixing plots would support our findings that DIC production can occur in the lower to outer estuary. We will include the DIC along the salinity gradient in May to August from 1997 to 2020 in the supplementary material to show that we do observe DIC production and consumption and include this in our results and discussion. As written in the submitted article on line 630 we believe a site-specific study would better explain the DIC consumption and production in this region.**

Fig.2a is difficult to read. Add the legend into the plot may be better.

**We will include the legend in the plot to make it clearer for the reader.**

L. 345 Why "*Such dominating heterotrophy in recent years (2018–2020) and DIC generation in the upper region (z1), would subsequently reduce the internal DIC load in the mid to lower estuary*"?

**We wanted to identify the difference between DIC in the upper and mid estuary would be lower because DIC was produced in the upper estuary and therefore higher concentrations in the upper region. A smaller difference between DIC concentration in the upper and mid-estuary indicates a smaller difference between DIC loads. Therefore, a lower internal DIC load in the mid to lower estuary.**

**To clarify we will revise this sentence to 'The dominating heterotrophy in recent years (2018–2020) and subsequent DIC generation in the upper region (z1) could reduce the difference between the DIC load in the upper and mid estuary. In turn, reducing the internal DIC load in the mid to lower estuary (Fig. 4).'**

Discussion:

L.420 According to this sentence, the author has identified a correlation between POC and DIC in another manuscript that is currently under review. I am curious about the differences between this manuscript and the one mentioned. The main focus of this manuscript appears to be the role of POC in the upper estuary in driving DIC increase in the mid and lower estuary. If this point has already been addressed in the author's other manuscript, it may not be appropriate to emphasize it again in this manuscript.

**Thank you for highlighting this. To clarify Rewrie et al. had only identified a significant increase in the annual mean DIC in the mid-lower estuary. Therefore, we will remove 'identified by Rewrie et al. (in review).'**

L.425 Please revise this sentence. It is not complete.

**Upon feedback from both reviewers, to clarify we will change this section from line 423 to 427 to 'The upper estuary POC in late spring and summer tripled since the onset of the recovery state in 1997, which we suggest is driven largely by allochtonous POC produced in the Elbe River and autochthonous POC produced in the upper estuary. Abril et al. (2002) reported that POC mineralisation efficiency (i.e., the percentage of POC mineralized) is a linear function of POC concentration, and considering the increased POC concentration, we can expect a higher turnover of POC in the estuary in recent years. That is, from 1997 to 2020, the increase in POC in the upper estuary enhanced the availability of POC for remineralisation, and subsequently increased DIC production, as was observed in the increase in DIC concentration in the Elbe Estuary (Fig 3).'**

L.430 I don't believe that the statement "*there are no other major sources of carbon along the estuary (Abril et al., 2002)*" can directly suggest that POC was efficiently remineralized and converted to DIC by the mid-estuary. The role of DOC in this process should also be considered. It is important to investigate the potential contribution of DOC to DIC through remineralization processes in the mid-estuary.

**As requested by both reviewers we would discuss the potential contribution of DOC by calculating the removal of OC for both DOC and POC (please find a further elaborated answer in the beginning of this response).**

---

## Author Response (AR3)

**Author Response to Minor Revision**

The authors have made a commendable effort in addressing the reviewer comments. The manuscript has been improved in a number of aspects, particularly in distinguishing the aims of this paper relative to a similar paper recently published by the same lead author. The manuscript has also been improved in providing more detail as to methodology. But with these additions, some further questions arise concerning the methods by which riverine inputs were derived and whether or not tidal exchange effects were considered (see below). The manuscript would benefit from some additional explanations regarding these issues.

**We thank the reviewer for their helpful review of the paper and we have attempted to carefully address the reviewer's comments in the below sections. Please note the line numbers refer to the final document and therefore not the tracked document.**

Methods

Section 2.7 (estimation of monthly loads): please specify how frequently DIC and POC are measured for riverine inputs. Also, prior work has shown that C concentrations vary with discharge (POC typically increasing, DIC decreasing), therefore concentration-discharge relationships are commonly used when estimating loads. Was that done here? Lastly, I am not seeing how you are taking into account the effects of tidal exchange on C concentrations in the estuary.

**The frequency of sampling was described in section 2.2 on lines 148-149. We edited the sentence to include 'once per month'. To clarify the 'Monthly mean POC and DOC' refers to mean of each zone we included 'for each zone in' on line 231 (text additions are italicized):**

| Lines | |
|---|---|
| **147-149** | Sampling was generally carried out *once per month* in February, May, June, July, August and November (see Rewrie et al. (2023) for a detailed description). |
| **231** | Monthly mean POC and DIC for *each zone in* May to August were calculated from 1997 to 2020, |

**We thank the reviewer for their suggestion to include concentration-discharge relationships. The reasons for calculating DIC and POC loads was to determine if the loads of these 2 parameters were the same magnitude, as well as to compare the internal DIC load during the non-drought and the drought period. This was described on lines 244-251 with 'The statistical differences between internal DIC loads during the recent drought (2014–2020) and non-drought (1997–2013) periods, and the differences between the internal DIC load in the mid-lower estuary and the POC load in the upper estuary (z1), for May to August were tested.' Therefore, we believe the application of concentration-discharge relationships would be beyond the scope for this specific paper.**

**As the sampling was conducted by helicopter, this permitted sampling at full ebb tide and this allowed the greatest possible synoptic comparability between the samples with regard to the influence of the tides. This means that all samples were collected at the same (or similar) tidal phase, i.e. during the ebb tide reflecting outflow from the estuary. This we consider important for consistency of the monthly sampling in this tidally**

**dominated system. To clarify this we have edited the following sentence (text additions are italicized):**

| Line | |
|------|---|
| **145-147** | The FGG Elbe took  surface water samples from 36 stations in the estuary (Fig. 1a) by helicopter at full ebb current, *which permitted the greatest possible synoptic comparability between the samples with regard to the influence of the tides  (ARGE Elbe, 2000).* |

**Monthly sampling unfortunately does not allow to resolve the tidal impacts (ebb vs flood tide) on carbon dynamics within the framework of this manuscript. Please also see the answer to the last reviewer comment in reference to this.**

An explanation of what is AOU and how it is calculated should appear in the main body of the paper given its central role in the author's assessment of changes in productivity.

**We thank the reviewer for this suggestion. We have included the AOU equation on line 156-162.**

Results

Lines 310-315: I am not understanding how your removal values are not rates (i.e., do not have a time component). Are you simply presenting the difference in concentration between two regions of the estuary as a percentage and calling this removal? Does this take into account changes in DOC and POC in the lower estuary due tidal exchange?

**As we followed the method in Amann et al. 2012 we are presenting the difference in percent and concentration. To clarify this we changed the following sentences (text additions are italicized):**

| Lines | |
|-------|---|
| **200-202** | To assess the remineralisation of the upper estuary POC and DOC in the Hamburg Harbour and the mid-estuary, *the percentage decrease in organic carbon ($OC_D$)* , was calculated according to the method used by Amann et al. (2012): |
| **207-209** | The *POC decrease, compared to zone 1*,  was *only* calculated for zone 2 and 3 due to the influence of the maximum turbidity zone in zone 4 and 5 (Amann et al. 2012). |
| **315-317** | *Compared to the upper estuary (z1),*  *the late spring and summer* DOC and POC decreased  *on average by 0.3 ± 21% and 40.6 ± 18%, respectively,* in Hamburg Harbour (z2-z3) and mid-estuary (*only* DOC in *z4-z5*), f*or the period 1997–2020 (Fig. 3).*  |
| **517-520** | There was on average 40.6 ± 18%  *decrease* of z1 POC , compared to *only* 0.3 ± 21%  *decrease* of z1 DOC in the estuary *(Fig. 3),* suggesting that heterotrophic respiration and DIC production was mainly fuelled by remineralisation of POC, which is in agreement with the findings of Amann et al. (2012). |

**On lines 208- 210 we described that we calculated the difference for zones 2-3 for POC and zones 2-5 for DOC and therefore did not calculate the difference for the lower estuary. This was done because the lower estuary was identified as a region, which is predominately influenced by OC produced in the German Bight (coastal North Sea), as discussed in the '4.2 Controls on inorganic carbon in the lower-outer estuary' section on lines 643-690.**

Lines 335-340: I am not following this argument. Are you saying that the increase in DIC was due to greater internal production of CO2? How does the TA:DIC ratio support that view?

**Yes. To clarify the point that the TA:DIC ratio can indicate 'the additional DIC input was in the form of pCO2' we changed the following sentence:**

| Lines | |
|---|---|
| **339-342** | In addition to this evidence, the ratio of TA to DIC can serve as an indicator of the source of carbon, *and specifically when < 1 this can reflect DIC input in the form of $CO_2$* (Joesoef et al., 2017)*, which was observed in this temperate estuary*. |

Lines 405-410: As per my earlier comment, how is the non-/conservative behavior of DIC (i.e., effects of tidal exchange) taken into account when estimating the internal load?

**Tidal exchange unfortunately cannot be resolved with the monthly samples we have used in this study. However, based on other studies with high-frequency time series (Cox et al. 2015; Voynova et al. 2015) and studies describing the net transport and loading of organic carbon (van Beusekom et al., 2009) we can deduce that tidal exchange is most likely an important modulator of organic carbon, especially in the lower and outer estuary, where organic carbon produced in the German Bight is an important factor (van Beusekom et al., 1999; Schulz et al., 2023). As discussed and referenced in the '4.2 Controls on inorganic carbon in the lower-outer estuary' section, specifically on lines 662-668 and 673-674.**

**The non-conservative behaviour of DIC in the estuary (z1-z5) has been accounted for in Figure 4 and in the SOM in Figures S4-S7 (lines 348-352 (results) and 665-668 (discussion)), as well as through the DIC:TA ratio (Fig. 4 and on lines 339-344 in the results).**

**Also, we would like highlight that in the submitted manuscript, we suggested that a follow-up study should concentrate on the nearshore waters, whereby both flood and ebb tide should be considered, on lines 761-763.**

**References**

**Cox, T. J., Maris, T., Soetaert, K., Kromkamp, J. C., Meire, P., & Meysman, F.: Estimating primary production from oxygen time series: A novel approach in the frequency domain. Limnology and Oceanography: Methods, 13(10), 529-552, https://doi.org/10.1016/j.seares.2008.06.005, 2015**

Schulz, G., Sanders, T., Voynova, Y. G., Bange, H. W., & Dähnke, K.: Seasonal variability of nitrous oxide concentrations and emissions in a temperate estuary. Biogeosciences, 20(15), 3229-3247, https://doi.org/10.5194/bg-20-3229-2023, 2023.

Van Beusekom, J. E. E., Brockmann, U. H., Hesse, K. J., Hickel, W., Poremba, K., & Tillmann, U.: The importance of sediments in the transformation and turnover of nutrients and organic matter in the Wadden Sea and German Bight. German Journal of Hydrography, 51(2-3), 245-266, DOI: 10.1007/BF02764176, 1999.

Van Beusekom, J. E., Loebl, M., & Martens, P.: Distant riverine nutrient supply and local temperature drive the long-term phytoplankton development in a temperate coastal basin. Journal of Sea Research, 61(1-2), 26-33, https://doi.org/10.1016/j.seares.2008.06.005, 2009.

Voynova, Y. G., Lebaron, K. C., Barnes, R. T., & Ullman, W. J.: In situ response of bay productivity to nutrient loading from a small tributary: The Delaware Bay-Murderkill Estuary tidally-coupled biogeochemical reactor. Estuarine, Coastal and Shelf Science, 160, 33-48, https://doi.org/10.1016/j.ecss.2015.03.027, 2015.

**Author Response to Major Revision**

**Response to both reviewers:**

**We thank both reviewers for their constructive and helpful review of our paper.**

**From both reviewers there were queries concerning the reference 'Rewrie et al. (in review)'. This manuscript was accepted by Limnology and Oceanography with the Article DOI: http://doi.org/10.1002/lno.12395 and is now published. We are happy to provide a brief explanation on the differences between the two papers.**

**Rewrie et al. (2023) assessed changes in DIC and ecosystem parameters from 1985 to 2018. From the abstract: 'Based on an extensive evaluation of key ecosystem variables, and an analysis of the available inorganic and organic carbon records, this study has identified three ecosystem states in recent history: the polluted (1985-1990), transitional (1991-1996) and recovery (1997-2018) states. The polluted state was characterised by very high dissolved inorganic carbon (DIC) and ammonium concentrations, toxic heavy metal levels, dissolved oxygen (DO) undersaturation and low pH. During the transitional state, heavy metal pollution decreased by > 50%, and primary production re-established in spring to summer, with weak seasonality in DIC. Since 1997, during the recovery state, DIC seasonality was driven by primary production, and DIC significantly increased by 11 $\mu$mol L$^{-1}$ yr$^{-1}$, and > 23 $\mu$mol L$^{-1}$ yr$^{-1}$ in the recent decade (2008-2018), in the mid to lower estuary, indicating that, along with the improvement in water quality the ecosystem state is still changing'**

**In the present manuscript, we focus on the changes in DIC in the recent recovery state (1997-2018) and extend the dataset by two years to 2020 due to more recent data availability. This study investigates the reason for the DIC increase over time by utilizing changes in organic carbon in the upper estuary. This study also evaluates the impact of the recent drought on the carbon cycling in the Elbe Estuary. We believe that publishing the manuscript described above, which precedes this one will answer most questions raised by the reviewers. However, we have attempted to carefully address the reviewer's comments in the sections below.**

**We would like to highlight that we have included a graphical abstract after the abstract. We would like to also highlight we have edited lines 707-711 to associate the decrease in DIC export load with the changes in seasonal internal DIC loads during the drought period.**

**Please find below the response to Referee_1 on pages 2-7 and Referee_2 on pages 8-14.**

Comments to Authors

This is an interesting dataset describing longitudinal gradients and long-term trends in POC and DIC for the Elbe Estuary. The findings from an analysis of these data would be of interest to estuarine ecologists and to the broader community studying the global C cycle. My main concern with the paper is that there are many aspects that are either not well explained, or insufficiently explained. This issue pervades key components of the paper including conceptualization of the study, description of methods and inferences made from the data.

Methods: the level of detail in explaining methods is highly uneven among the various components of the study. For example, the means for calculating air-water CO2 fluxes is highly detailed, whereas other equally important elements (e.g., POC loads and DIC mineralization) are hardly described at all in the main body of the paper. These fluxes, and how they are derived should be explained more fully and at an early stage. For example, in the Abstract there is reference to the "spring internal DIC load" (line 30), but at this stage, the leader is not likely to understand what this is (remineralization of organic C) or how it was determined. Also, I did not see an explanation of how POC was measured (perhaps I missed this).

**To ensure the methods are clearly provided we have moved equations S2 to S5 from the supplementary materials into the methods section (lines 224-237). We have also included the equation on line 216 for the difference in DIC concentration between the mid, lower and outer estuary ($z4$–$z7$) relative to the upper region ($z1$).**

**We had referenced Rewrie et al. (2023) which extensively describes the inorganic and organic carbon measurement methods to reduce repetition. However, to increase clarity in this manuscript, we have included a sentence from line 151 to describe that POC was calculated as the difference between TOC and DOC (Table S1), with an estimated uncertainty of 20% based on the Pythagorean Theorem (U. Wiegel, pers. comm).**

Introduction: the first paragraph focuses on C cycling in estuaries, with a mention of eutrophication. The second paragraph focuses of climate change, and specifically, the occurrence of drought. The third (final) paragraph does not link the ideas presented in the first and second paragraphs in such a way as to provide a clear direction for the paper. There should be some consideration (prediction) of how estuarine C cycles may be affected by drought. My initial reaction was that drought would reduce external (watershed) inputs of POC and DOC to the estuary. I was surprised that there was no consideration here, or elsewhere, of the importance of allochthonous organic matter inputs, particularly as this is the driving mechanism accounting for the excess of C mineralization relative to autotrophic fixation in estuaries (see e.g., Hoellein et al. 2013). The third paragraph should provide some clear expectations of the direction of the paper from a conceptual point of view, which the stated objectives fail to do. Also, there should be some consideration of allochthonous C inputs and how these may have changed over time (and in response to drought).

**Thank you for this suggestion and reference. We do in fact argue in the later sections of the manuscript that allochtonous POC sources are key to the DIC processing in the estuary. To highlight the importance of allochthonous C in the introduction, we changed**

the first paragraph from line 55: 'River-borne and in situ primary production supplies allochthonous and autochthonous organic carbon to and within estuaries (Abril et al. 2002; Hoellein et al. 2013), subsequently providing labile forms of carbon.'

In the second paragraph, we described the influences of extreme floods and droughts on OM and nutrient cycling within estuaries on lines 74-82. To link the first and second paragraph we changed the sentence from line 79: 'This in turn can extend the retention of carbon and nutrients during droughts, permitting more extensive remineralization of allochthonous and autochthonous OM within an estuary (Hitchcock and Mitrovic, 2015), and subsequently altering carbon and nutrient cycling'

To clarify the reasons behind assessing the period between 1997 and 2020 we changed the third paragraph on lines 92-94 to 'To assess the impact of the drought on carbon cycling in the estuary, we use a longer period between 1997 and 2020, to allow comparisons between non-drought and drought periods'

The reliance on Rewrie et al. in review, here, and at many points throughout the paper, is not helpful, particularly as these are vague references to "ecosystem recovery", "major shifts in ecosystem state" and "amelioration of water quality". It is not clear to the readers of this paper what these changes are, and what implications they may have for C cycling in the Elbe. There is a subsequent statement that these changes include a reduction in BOD and an increase in NPP. The former implies that the changes may have to do with improved wastewater treatment resulting in reduced organic matter inputs to the estuary. But if that is the case, should these fluxes not be accounted for in this paper describing trends in POC and DIC? Also, if wastewater treatment practices have been improved, this should bring about a reduction in nutrient inputs, and potentially diminish, not enhance NPP. I short, I found it difficult to understand the long-term trends presented in this paper while not understanding what are the changes occurring in this system that seem to be the focus of a different paper.

To clarify the key findings of Rewrie et al. (2023) we changed the introduction from line 94 and removed 'It has been described as the ecosystem recovery state of the estuary (Rewrie et al., 2023), following major shifts in the ecosystem state after the 1980s heavy pollution.' And included 'Since 1997, the ecosystem of the Elbe Estuary was designated to be in a recovery state, after heavy pollution in the 1980s, and a transitional state in the 1990s (Rewrie et al., 2023). The authors characterised the current recovery ecosystem state by non-toxic levels of heavy metals, which permitted the reestablishment of autotrophic and heterotrophic processes and OM cycling within the estuary'.

To clarify the changes in the water quality in the Elbe Estuary we specified the BOD changes observed in the upper estuary. Since the decrease was not directly inverse to the POC increase, we removed 'opposite of the observed POC increase' and included on line 481 'The summer mean BOD7 decreased from 12 ± 1.7 mg L-1 in 1997–2005 to 8 ± 1.1 mg L-1 in 2006–2020'. After this sentence we also included 'While there was a continuous decrease in nutrients from the late 1990s (Wachholz et al. 2022), the nutrient supply was still sufficient to support phytoplankton production (Kamjunke et al., 2021; Dähnke et al 2022).' This highlights that despite the overall decrease in nutrients the concentrations were predominately sufficient for primary production in the Elbe River and Estuary, as has been found by other researchers (e.g. Dähnke et al 2022), who will also be referenced.

**Dähnke, K., Sanders, T., Voynova, Y., & Wankel, S. D. (2022). Nitrogen isotopes reveal a particulate-matter-driven biogeochemical reactor in a temperate estuary. Biogeosciences, 19(24), 5879-5891.**

**Wachholz, A., Jawitz, J. W., Büttner, O., Jomaa, S., Merz, R., Yang, S., & Borchardt, D. (2022). Drivers of multi-decadal nitrate regime shifts in a large European catchment. Environmental Research Letters, 17(6), 064039.**

**We would like to highlight that in the submitted manuscript we had indeed identified a potential reduction in nutrient inputs in 2018 to 2019 on lines 617-619, which was associated with the drought.**

Results: throughout the paper, loads are presented as mass per unit of time (e.g., the total mass of CO2 leaving the estuary), which is not very helpful to facilitating cross-system comparisons (vs. presenting these as values per unit area of the estuary). By analogy, river loads (watershed export) are more commonly normalized to watershed area (i.e., as a yield per square meter) to allow comparisons among watersheds of different size. Readers could take the values provided in this paper and divide by the specified area of the estuary to obtain estimates for comparisons to other estuaries. But the potential for making inter-system comparisons would be enhanced if the authors were to present their data as per unit area of the estuary.

**Thank you for this suggestion. The reason why we used the mass per unit of time was to allow us to calculate a ratio of the export and compare to other estuaries on lines 692-706. To allow the reader to see our results in both forms of units we combined Gmol C yr$^{-1}$ to one axis with a log scale and have presented on the twin axis the export plot per unit area (see Figure 6 on line 465). We would like to highlight that in the submitted manuscript we had compared the air-water CO2 flux range with flux estimates for North Sea tidal estuaries determined by Volta et al. (2016) from line 695.**

Other Comments:

The Abstract lacks a 'big picture' perspective. The overall findings of the study are difficult to discern among the details of the results.

**To put our results in perspective to other studies we changed the abstract on lines 37-42: 'We have identified that seasonal changes in DIC processing in an estuary require consideration when estimating both the long-term and future changes in air-water CO2 flux and DIC export to coastal waters. Regional and global carbon budgets should therefore take into account carbon cycling estimates in estuaries, and their changes over time in relation to impacts of water quality changes and extreme hydrological events.'**

Site Description: it would be helpful if this included an indication of salinity levels along the length of the estuary (perhaps add data to Figure 1, or at least delineate polyhaline, mesohaline, etc.).

**Thank you for this suggestion. To address this we included an average with standard deviation salinity gradient on Figure 1b.**

DOC: I was surprised that in a paper on estuarine C dynamics there was virtually no mention of DOC. Is it the case that internal production of POC and subsequent remineralization of

POC are the dominant C fluxes in this system?  At a minimum, it would be useful to report the proportions of total C represented by DIC, POC and DOC in river inputs to the estuary vs. relative contributions in export to the sea (and for drought vs. non-drought conditions).

**Yes, we suggest that the production of allochtonous POC (in the river and in the coastal regions adjacent to the estuary) and its subsequent remineralization of POC are controlling the C fluxes in this system, and are responsible for the recent observed increase in DIC. We included a reference in the introduction on lines 104-107 describing the respiration of organic carbon in the Elbe Estuary: 'The organic carbon cycling in the upper Elbe Estuary was evaluated before (Amann et al., 2012), and the study identified that from the late 1990s the POC fraction fueled heterotrophic respiration in the estuary, whereas the removal of DOC was negligible. However, the last decade was not included in this analysis'**

**In response to both reviewers, we included information on DOC to show the concurrent DOC changes as well, to support our findings that the remineralization of POC, rather than DOC fuels the DIC production. To address the reviewer's concern we estimated the removal of DOC and POC in the estuary, defined as filtering capacity in Amann et al. (2012) in the Elbe Estuary.**

**To assess the importance of the upper estuary DOC and POC respiration and subsequent DIC production in the Hamburg Harbour and the mid-estuary the organic carbon removal estimated as the percentage decrease in OC, as shown in Amann et al. (2012), was calculated. Please refer to section 2.5 Remineralisation of upper estuary POC and DOC on lines 192 to 204.**

**We found that 'The mean DOC and POC removal in Hamburg Harbour and mid-estuary (DOC in mid-estuary), estimated as the OC percentage decrease in zones 2 to 5 (DOC in zones 4 and 5) from the initial value in upper estuary (z1), was 0.3 ± 21% and 40.6 ± 18%, respectively (Fig. 3), in late spring and summer from 1997 to 2020. This corresponded to a mean concentration decrease of 10 ± 69 µmol kg-1 and 157 ± 106 µmol kg-1 in DOC and POC. This indicates that respiration of upper estuary POC, rather than DOC, dominates heterotrophic activity in the Hamburg Harbour and mid Elbe Estuary, and subsequent DIC production.' This was included on lines 309-315.**

**To integrate our findings into the discussion we included on lines 509-512: 'There was on average 40.6 ± 18% removal of z1 POC (Fig. 3), compared to 0.3 ± 21% removal of z1 DOC in the estuary, suggesting that heterotrophic respiration and DIC production was mainly fuelled by remineralisation of POC, which is in agreement with the findings of Amann et al. (2012).'**

**To further support upper estuary POC dominates DIC production in the mid-estuary, we would like to highlight that in the submitted article we discussed (on lines 508-509) 'Moreover, the magnitude of along-estuary DIC gain in the mid-estuary and POC input into the estuary show no significant difference in late spring and summer (Table S5).' We would also like to highlight in the submitted article we discussed POC in % of SPM was used to describe the mineralisation of POC in the mid-estuary. From line 516 'We find that POC drops to < 4% of SPM in May to August (1997–2020) in the mid estuary (z4–z5, Fig. S9), indicating widespread OM remineralization in the estuary.'**

We thank the reviewers for the suggestion to include remineralisation of DOC to strengthen the story that the remineralisation of upper estuary POC dominates DIC production in the Hamburg Harbour to the mid-estuary.

In the lower to outer estuary, we have identified that other sources of OM likely support DIC production as described in the submitted article on lines 636 to 682. To assess DOC production in this region, we assessed the mixing of DOC along salinity for May to August between 1997 and 2020. We included on lines 643-466: 'We observed mainly positive (42%), and few negative (13%), deviations of DOC from conservative mixing in May to August between 1997 and 2020 (Figs. S10–S13), also suggesting an addition of OC in the lower and outer region.' This corroborates our suggestion of OC production in the outer estuary can fuel DIC production therein. We included additional DOC mixing along the salinity gradient figures in the supplementary material (Figs. S10-S13) to support our findings in the discussion.

Amann, T., Weiss, A., & Hartmann, J. (2012). Carbon dynamics in the freshwater part of the Elbe estuary, Germany: Implications of improving water quality. Estuarine, Coastal and Shelf Science, 107, 112-121.

Results (line 240): there is frequent use of indirect metrics (AOU, pH) to make inferences about autotrophic activity. Are there no primary data that can be used to support these inferences (e.g., CHLa measurements)?

To support primary production in spring and summer we plotted a seasonal plot of the monthly mean chl-a at 585.5 Elbe-km calculated from 1997 to 2020. We placed this plot in the supplementary material (Figure S2), since it is only supporting the findings of AOU and pH in the upper estuary, without altering the manuscript findings. AOU we find is a more powerful measurement of production/respiration balance. A time-series of chl-a does not provide more evidence than POC, which was chosen to represent the labile carbon produced during primary production and available for remineralisation, and for which we found a strong and significant correlation with DIC, in the same order of magnitude, and already in units of carbon. We included (in italics) on line 288-290 'This suggests that dominating autotrophy depletes DIC in the upper estuary, and most likely the upstream river regions, *which is supported by highest seasonal chlorophyll a concentrations in May to August at 585.5 Elbe-km, reaching 166 ± 74 µg L-1 (Fig. S2).'*

Results (line 260): the statement "significant POC increases occurred..." is followed by some specified values, but it is unclear what these numbers represent (the mean concentration? the increase in concentration? If the latter, increase relative to what?).

To clarify this sentence we changed it to 'Significant mean POC increases occurred in late spring (May, 14 µmol C kg-1 yr-1) and summer (June–August, 8 µmol C kg-1 yr-1) in the upper estuary (Fig. 4, Table 1) from 1997 to 2020'.

Results (line 278): I did not understand why the TA:DIC ratio should be of interest, or what is the significance of this ratio being <1.

To clarify the importance of using the TA: DIC ratio we included the following reference on lines 334-336: 'In addition to this evidence, the ratio of TA to DIC can serve as an indicator of the source of carbon (Joesoef et al., 2017).'

Discussion (line 425): do you mean to say that mineralization rates increase linearly with POC concentrations, or that mineralization efficiency increases (i.e., that the proportion of POC that is remineralized increases)?

**Upon feedback from both reviewers, to clarify we changed this section from lines 497-505 to 'The upper estuary POC in late spring and summer tripled since the onset of the recovery state in 1997, which we suggest is driven largely by allochtonous POC produced in the Elbe River and autochthonous POC produced in the upper estuary. Abril et al. (2002) reported that POC mineralisation efficiency (i.e., the percentage of POC mineralized) is a linear function of POC concentration, and considering the increased POC concentration, we can expect a higher turnover of POC in the estuary in recent years. That is, from 1997 to 2020, the increase in POC in the upper estuary enhanced the availability of POC for remineralisation in the Elbe Estuary, and subsequently increased DIC production, as evidenced by the concomitant increase in DIC concentration (Fig. 4).'**

Table 3.2: it would be helpful to include the standard error of the slope.

**We included the standard error of the slope for DIC and TA in Table 1.**

Figure 4: it is somewhat confusing to use the designation "m-1" as this is much more commonly used to indicate per meter. Perhaps "mon-1" would be the better abbreviation for monthly values?

**Thank you for highlighting this. We changed the unit to $mon^{-1}$ in the figure and $month^{-1}$ in text.**

Rewrie et al. (2023) investigated the variations in biogeochemical parameters, including DIC, POC, DO, and pH, in the Elbe Estuary over a long time period of 23 years. They discovered a significant long-term increase in DIC in the Elbe Estuary, which might be primarily influenced by a rise in POC content in the upper estuary. The researchers suggest that the increased internal load of DIC during the drought period can be attributed to an extended residence time in the estuary, allowing for a longer period of POC remineralization. The manuscript highlights the correlation between POC in the upper estuary and DIC in the mid and lower estuary, which may provide valuable insights for understanding DIC variations driven by climate change and human-induced disturbances. However, I have several concerns regarding the manuscript, and I hope the authors will carefully consider them.

Main comments

1.The authors have solely examined the correlation between POC data in the upper estuary and DIC concentrations in the mid and lower estuary. Including POC data from the mid and lower estuary in the manuscript would offer a clearer understanding of the relationship between POC and DIC. Additionally, while the authors have provided a preliminary estimate of inorganic carbon export, they have not thoroughly discussed other potential factors influencing inorganic carbon export. I am curious about the relationship between dissolved organic carbon (DOC) and DIC in the Elbe Estuary. Since labile DOC can also be consumed by microbes, it may contribute to the DIC pool. I recommend the authors add some discussions on the correlations between DOC and DIC.

**We thank the reviewer for this suggestion to include DOC.**

**We included a reference in the introduction on lines 104-107 describing the respiration of organic carbon in the Elbe Estuary: 'The organic carbon cycling in the upper Elbe Estuary was evaluated before (Amann et al., 2012), and the study identified that from the late 1990s the POC fraction fueled heterotrophic respiration in the estuary, whereas the removal of DOC was negligible. However, the last decade was not included in this analysis'**

**In response to both reviewers, we included information on DOC to show the concurrent DOC changes as well, to support our findings that the remineralization of POC, rather than DOC fuels the DIC production. To address the reviewer's concern we estimated the removal of DOC and POC in the estuary, defined as filtering capacity in Amann et al. (2012) in the Elbe Estuary.**

**To assess the importance of the upper estuary DOC and POC respiration and subsequent DIC production in the Hamburg Harbour and the mid-estuary the organic carbon removal estimated as the percentage decrease in OC, as shown in Amann et al. (2012), was calculated. Please refer to section 2.5 Remineralisation of upper estuary POC and DOC on lines 192 to 204.**

**We found that 'The mean DOC and POC removal in Hamburg Harbour and mid-estuary (DOC in mid-estuary), estimated as the OC percentage decrease in zones 2 to 5 (DOC in zones 4 and 5) from the initial value in upper estuary (z1), was 0.3 ± 21% and**

**40.6 ± 18%, respectively (Fig. 3), in late spring and summer from 1997 to 2020. This corresponded to a mean concentration decrease of 10 ± 69 μmol kg-1 and 157 ± 106 μmol kg-1 in DOC and POC. This indicates that respiration of upper estuary POC, rather than DOC, dominates heterotrophic activity in the Hamburg Harbour and mid Elbe Estuary, and subsequent DIC production.' This was included on lines 309-315.**

**To integrate our findings into the discussion we included on lines 509-512: 'There was on average 40.6 ± 18% removal of z1 POC (Fig. 3), compared to 0.3 ± 21% removal of z1 DOC in the estuary, suggesting that heterotrophic respiration and DIC production was mainly fuelled by remineralisation of POC, which is in agreement with the findings of Amann et al. (2012).'**

**To further support upper estuary POC dominates DIC production in the mid-estuary, we would like to highlight that in the submitted article we discussed (on lines 508-509) 'Moreover, the magnitude of along-estuary DIC gain in the mid-estuary and POC input into the estuary show no significant difference in late spring and summer (Table S5).' We would also like to highlight in the submitted article we discussed POC in % of SPM was used to describe the mineralisation of POC in the mid-estuary. From line 516 'We find that POC drops to < 4% of SPM in May to August (1997–2020) in the mid estuary (z4–z5, Fig. S9), indicating widespread OM remineralization in the estuary.'**

**We thank the reviewers for the suggestion to include remineralisation of DOC to strengthen the story that the remineralisation of upper estuary POC dominates DIC production in the Hamburg Harbour to the mid-estuary.**

**In the lower to outer estuary, we have identified that other sources of OM likely support DIC production as described in the submitted article on lines 636 to 682. To assess DOC production in this region, we assessed the mixing of DOC along salinity for May to August between 1997 and 2020. We included on lines 643-466: 'We observed mainly positive (42%), and few negative (13%), deviations of DOC from conservative mixing in May to August between 1997 and 2020 (Figs. S10–S13), also suggesting an addition of OC in the lower and outer region.' This corroborates our suggestion of OC production in the outer estuary can fuel DIC production therein. We included additional DOC mixing along the salinity gradient figures in the supplementary material (Figs. S10-S13) to support our findings in the discussion.**

**Amann, T., Weiss, A., & Hartmann, J. (2012). Carbon dynamics in the freshwater part of the Elbe estuary, Germany: Implications of improving water quality. Estuarine, Coastal and Shelf Science, 107, 112-121.**

2. The authors utilized the CO2SYS program to calculate TA and pCO2 using DIC and pH as input parameters. However, it should be noted that the TA model in the CO2SYS program does not account for the contribution of organic alkalinity. Previous studies have reported significant concentrations of organic alkalinity (ranging from 10 to 70 umol kg$^{-1}$) in rivers and estuaries. Ignoring the presence of organic alkalinity in the calculated TA values may introduce substantial uncertainties. Therefore, I recommend that the authors include discussions addressing the uncertainties associated with the calculated TA values.

**Thank you for this recommendation. We included a discussion to compare the calculated TA and the available DOC measurements (from FGG Elbe data portal) with previous studies that describe organic alkalinity. Please refer to lines 571-591: 'The**

calculated TA measurements could be influenced by the variability of OM (Kim et al., 2006; Kuliński et al., 2014). The CO2SYS program does not account for contribution of organic alkalinity in the calculated TA. While organic alkalinity typically constitutes a smaller fraction of TA compared to that provided by the inorganic compounds, it could still be a significant component of TA in systems influenced by dissolved OM inputs (Kuliński et al., 2014). For instance, Hunt et al. (2011) reported a 21-100% contribution of organic alkalinity to TA in 15 rivers of northern New England (USA) and New Brunswick (Canada), with low TA (116 µM to 956 µM) and extremely high DOC, up to 1480 µmol L-1. In the Vistula and Oder Rivers, characterised by an average TA and DOC concentrations of 2965 ± 568 µmol kg-1 and 560 ± 77 µmol kg-1, organic TA contributed < 8% (Kuliński et al., 2014). Kuliński et al. (2014) reported that the higher percentage of organic alkalinity contributing to TA in the rivers investigated by Hunt et al. (2011) was due to the lower amount of TA. Mean DOC and TA in the Elbe Estuary (z1-z7) were respectively 498 ± 92 µmol kg-1 and 1985 ± 309 µmol kg-1 for the entire record (1997-2020, not shown). Compared to the Elbe Estuary, similar TA (< 2200 µmol kg-1) and DOC (< 450 µmol kg-1) concentrations were observed in an intertidal saltmarsh in the northeast USA, where the organic alkalinity fraction contributed a minimal amount of 0.9-4.3% to the TA (Song et al., 2020). Thus, the organic alkalinity could constitute only a small fraction of the TA in the Elbe Estuary. In future studies, either the difference between calculated and measured TA (Kuliński et al., 2014) or direct measurements of organic alkalinity (Song et al. 2020) could be used to quantify the organic alkalinity influence on TA in the Elbe Estuary.'**

3.This manuscript extensively references a paper (Rewrie et al., in review) that is currently under review. It is generally not considered appropriate to rely heavily on the findings of a paper that is still undergoing the review process. Additionally, it seems there is a high correlation between that paper and the current manuscript. It would be helpful if the authors could provide an introduction outlining the differences and commonalities between this manuscript and the paper they have cited (Rewrie et al., in review) in the response to reviewers.

**We have explained the Rewrie et al. (2023) manuscript above.**

**To clarify the key findings of Rewrie et al. (2023) we changed the introduction from line 94 and removed 'It has been described as the ecosystem recovery state of the estuary (Rewrie et al., 2023), following major shifts in the ecosystem state after the 1980s heavy pollution.' And included 'Since 1997, the ecosystem of the Elbe Estuary was designated to be in a recovery state, after heavy pollution in the 1980s, and a transitional state in the 1990s (Rewrie et al., 2023). The authors characterised the current recovery ecosystem state by non-toxic levels of heavy metals, which permitted the reestablishment of autotrophic and heterotrophic processes and OM cycling within the estuary'.**

**We changed the second aim on lines 109-110 to be specific '(2) investigate how the onset of the recent drought has modulated the carbon cycling within the estuary.'**

4.The authors have utilized potentiometric pH and DIC measurements to calculate pCO2. However, it is important to note that potentiometric pH measurements may introduce significant uncertainties when measuring pH in saline waters. What is the salinity of the samples collected in the outer estuary? The authors should take into consideration this potential source of uncertainty in their analysis and discussion.

**On lines 180, we applied the recommended total standard uncertainty for pH of 0.01 units. This error was also suggested in Dickson (1993). We included this reference as well on line 181.**

**Dickson, A. G. (1993). The measurement of sea water pH. Marine chemistry, 44(2-4), 131-142.**

Minor comments

Abstract:

L.25 Is it possible that a portion of the gained DIC can be attributed to the degradation of DOC?

**We believe DIC gained by respiration of DOC is negligible compared to POC remineralization as answered above.**

Introduction:

L.55 "source"?

**Thank you for highlighting this. We will change this to 'source'.**

L.45-60 Why do the authors use "OM" and "POC" interchangeably in this paragraph? Does "OM" include DOC?

**Our aim was not to use OM and POC interchangeably. We wanted to highlight that organic carbon was a form of organic material. To clarify we are referring to organic carbon in the paragraph and upon feedback from the Referee_1, we changed the sentence from line 55: 'River-borne and in situ primary production supplies allochthonous and autochthonous organic carbon to and within estuaries (Abril et al. 2002; Hoellein et al. 2013), subsequently providing labile forms of carbon.'**

**We changed the sentence on line 62 to 'This reduces labile OM export to the adjacent coastal waters (Abril et al. 2002; Crump et al., 2017; Sanders et al., 2018)'.**

L.55 I suggest that the authors include an introduction discussing the contribution of DOC degradation to DIC.

**We included a reference in the introduction describing the respiration of organic carbon in the Elbe Estuary as answered above (and see lines 104-107).**

Method:

L.150 Please add a reference for the CO2SYS program.

**We included the reference (Lewis and Wallace, 1998) after 'Using the CO2SYS program version 2.5 in Excel'.**

L.130 what is the precision of pH, DOC, and DIC measurements?

**We provided the significant digits/indication of results in Table S1.**

L.190 "from May to August"?

**The correction factor was related to the monthly river discharge. To clarify, we changed the sentence to 'A correction factor to the monthly river discharge was applied to each estuarine zone (zones 1 to 6) to account for tributary inputs along the estuary (Amann et al., 2015).'**

L 220 In the study of an estuary, why did the authors use a Schmidt number (Sc) in freshwater? The gas transfer velocity parameter (k) described by Wanninkhof (2014) is generally applicable to the open ocean.

**The flux of $CO_2$ between water and atmosphere was calculated for zones 1 to 6. For the entire record (1997-2020) the mean salinity in zone 6 was 11 ± 5 psu and for zones 1 to 6 was 2 ± 4 psu. In Wanninkhof (2014) the coefficients for the Schmidt number for seawater is for 35 psu. Therefore, due to the relatively lower salinity range in the Elbe Estuary we applied the freshwater Schmidt number. We decided to use a Schmidt number (Sc) in freshwater to also allow for comparisons of the calculated flux of $CO_2$ between water and atmosphere with published studies on the Elbe Estuary (Norbisrath et al. (2022) and Amann et al. (2015)).**

**We did not compare different approaches to calculate k as the aim was to provide a provisional evaluation of the overall inorganic carbon export as stated with the section title of '4.3 Tentative inorganic carbon export estimates' on line 684. We also compared our calculated $CO_2$ flux estimates to those calculated in Norbisrath et al (2022), where the authors did not use a correction factor for the k calculation for coastal waters, and we found similar estimates (please refer to lines 271-272). Even though we have not used a correction factor for the k calculation (such as in Volta et al. 2016), the mean air-water $CO_2$ flux (21 ± 6 mol C m$^{-2}$ yr$^{-1}$, 1997-2020) places the Elbe Estuary within flux estimates for North Sea tidal estuaries and specifically the Elbe Estuary of 27.4 mol C m$^{-2}$ yr$^{-1}$ (Volta et al. 2016). This is highlighted on lines 695-696. As requested by the other reviewer we included the air-water $CO_2$ flux in both mol C m$^{-2}$ yr$^{-1}$ and Gmol yr$^{-1}$ in Figure 6.**

Result:

L.285 How does the physical mixing between freshwater and seawater influence DIC in the outer estuary?

**In May to August, we found positive and negative non-conservative mixing in DIC along the salinity gradient in 43% and 23% of the assessed months between 1997 and 2020. This was not included in the manuscript however, we acknowledge that the non-conservative mixing plots would support our findings that DIC production can occur in the lower to outer estuary. We included the DIC along the salinity gradient in May to August from 1997 to 2020 in the supplementary material in Figs. S4-S7. We integrated these results on lines 342-346: 'The DIC variability along the salinity gradient, with conservative mixing line between the river and North Sea end members, showed non-conservative behaviour, i.e. positive (43%) and negative (23%) deviations from the mixing line in May to August (Figs. S4-S7). The positive deviations indicate mainly an internal source of DIC in the lower to outer estuary.' And in the discussion on lines 658-**

**661: 'This potential source of DIC from marine OC in the lower to outer estuary was also indicated by predominate positive excursions in DIC from conservative mixing lines in May to August between 1997 and 2020 (Figs. S4–S7), and suggested by Schulz et al. (2023).'**

**As written in the submitted article on lines 753-755 we believe a site-specific study would better explain the DIC consumption and production in this region.**

Fig.2a is difficult to read. Add the legend into the plot may be better.

**We included the legend in the plot to make it clearer for the reader.**

L. 345 Why "*Such dominating heterotrophy in recent years (2018–2020) and DIC generation in the upper region (z1), would subsequently reduce the internal DIC load in the mid to lower estuary*"?

**We wanted to identify the difference between DIC in the upper and mid estuary would be lower because DIC was produced in the upper estuary and therefore higher concentrations in the upper region. A smaller difference between DIC concentration in the upper and mid-estuary indicates a smaller difference between DIC loads. Therefore, a lower internal DIC load in the mid to lower estuary.**

**To clarify revised this sentence to 'That is, the dominating heterotrophy in recent years (2018–2020) and subsequent DIC generation in the upper region (z1) could reduce the DIC load difference between the upper and mid estuary. In turn, this reduces the internal DIC load in the mid estuary (Fig. 5).'**

Discussion:

L.420 According to this sentence, the author has identified a correlation between POC and DIC in another manuscript that is currently under review. I am curious about the differences between this manuscript and the one mentioned. The main focus of this manuscript appears to be the role of POC in the upper estuary in driving DIC increase in the mid and lower estuary. If this point has already been addressed in the author's other manuscript, it may not be appropriate to emphasize it again in this manuscript.

**Thank you for highlighting this. To clarify Rewrie et al. had only identified a significant increase in the annual mean DIC in the mid-lower estuary. Therefore, we removed 'identified by Rewrie et al. (in review).'**

L.425 Please revise this sentence. It is not complete.

**Upon feedback from both reviewers, to clarify we changed this section from lines 497-505 to 'The upper estuary POC in late spring and summer tripled since the onset of the recovery state in 1997, which we suggest is driven largely by allochtonous POC produced in the Elbe River and autochthonous POC produced in the upper estuary. Abril et al. (2002) reported that POC mineralisation efficiency (i.e., the percentage of POC mineralized) is a linear function of POC concentration, and considering the increased POC concentration, we can expect a higher turnover of POC in the estuary in recent years. That is, from 1997 to 2020, the increase in POC in the upper estuary enhanced the availability of POC for remineralisation in the Elbe Estuary, and**

**subsequently increased DIC production, as evidenced by the concomitant increase in DIC concentration (Fig. 4).'**

L.430 I don't believe that the statement "*there are no other major sources of carbon along the estuary (Abril et al., 2002)*" can directly suggest that POC was efficiently remineralized and converted to DIC by the mid-estuary. The role of DOC in this process should also be considered. It is important to investigate the potential contribution of DOC to DIC through remineralization processes in the mid-estuary.

**As requested by both reviewers we would discuss the potential contribution of DOC by calculating the removal of OC for both DOC and POC (please find a further elaborated answer in the beginning of this response).**